# Robust Prediction-Powered Inference under Data Corruption

## ABSTRACT

This paper proposes a robust prediction-powered semi-supervised statistical learning and inference framework. Existing prediction-powered inference (PPI) methods use pre-trained machine-learning models to impute unlabeled samples and calibrated the imputation bias, based on the assumption of covariate homogeneity between the labeled and unlabeled datasets. However, violation of the homogeneity assumption, such as distribution shifts and data corruption, can undermine the effectiveness of semi-supervised approaches and even break down the learning process. In response, we introduce robust estimation techniques to the imputation-and-then-calibration procedure of PPI. The approach can be easily integrated with general PPI methods and improves the robustness of them against the heterogeneity and the corruption in the unlabeled set. To make full use of the labeled and unlabeled data, a cross-validation procedure is also developed for selecting the shift/contamination level. Theoretical analysis shows that our method is consistent and robust under mild conditions. Numerical simulations and real-data applications also demonstrate the robustness and superiority of the proposed method.

## 1 INTRODUCTION

In statistical analysis, it is common to assume that the samples under examination have been collected in an appropriate and representative manner, and adhere to certain homogeneous patterns. This foundational assumption simplifies the analytical process and allows for the application of a wide range of statistical techniques. However, data in the real world often diverges significantly from these ideal conditions. Real-world data can be messy, incomplete, and inconsistent, presenting numerous challenges that can undermine the validity and reliability of statistical tools.

Data scientists often struggle with the scarcity of labeled data (Chapelle et al., 2009; van Engelen & Hoos, 2020). Data labeling can be both time-consuming and costly, often requiring extensive manual effort and expertise, or difficult due to privacy concerns. To address this, the semi-supervised learning (SSL) paradigm is well-developed to improve the performance of supervised learning by unlabeled data. Pseudo-label imputation using modern black-box machine learning models is a prevailing SSL method (Lee et al., 2013; Zhang et al., 2021; Hu et al., 2022; Sportisse et al., 2023). However, the direct use of these pseudo labels in model training will introduce endogenous bias, negatively affect the trained models and the downstream statistical inference tasks. Recently, the *prediction-and-then-calibration* procedure, which calibrates the imputation bias of unlabeled data under specific statistical models, has been studied by statisticians Chakrabortty & Cai (2018); Zhang et al. (2019); Zhang & Bradic (2022); Azriel et al. (2022). The calibration ensures reliable statistical estimation and inference in semi-supervised tasks. Based on these achievements, Angelopoulos et al. (2023a) proposes *prediction-powered inference* (PPI), a general framework that embeds predictions from *any* black-box machine learning model into valid statistical inference.

A critical bottleneck in deploying semi-supervised inference in real-world workflows is the assumption of distributional homogeneity between labeled and unlabeled data. In practice, labeled datasets are often 'gold standards'—carefully curated, cleaned, and representative of the target population. In contrast, unlabeled datasets are typically voluminous but collected 'in the wild,' making them susceptible to distribution shifts and Out-of-Distribution (OOD) noise (Koh et al., 2021). For instance, unlabeled data typically arises from different time periods, geographical locations, or data-collection pipelines than the labeled set, introducing covariate shifts and naturally occurring corruptions (Diakonikolas & Kane, 2023). Standard PPI methods rely on the strict assumption that labeled and

unlabeled data share the exact same distribution ($P_X$). When this assumption is violated, as is common in OOD generalization tasks, the bias correction mechanisms in PPI fail, leading to invalid confidence intervals and erroneous conclusions (Kluger et al., 2025). In this work, we model these distributional mismatches through the lens of the Huber contamination model. This allows us to develop a framework that is robust not only to adversarial outliers but, more importantly, to the inherent noise and distribution shifts prevalent in modern large-scale semi-supervised learning tasks. Figure 1 illustrates an example of gene expression level estimation under the SSL setting, using the sequence-to-expression transformer model trained in Vaishnav et al. (2022) for the label imputation. As the contamination proportion $\epsilon$ increases, the estimation error of PPI-type methods rises rapidly. Moreover, the coverage rate of the constructed confidence interval (CI) quickly deteriorates.

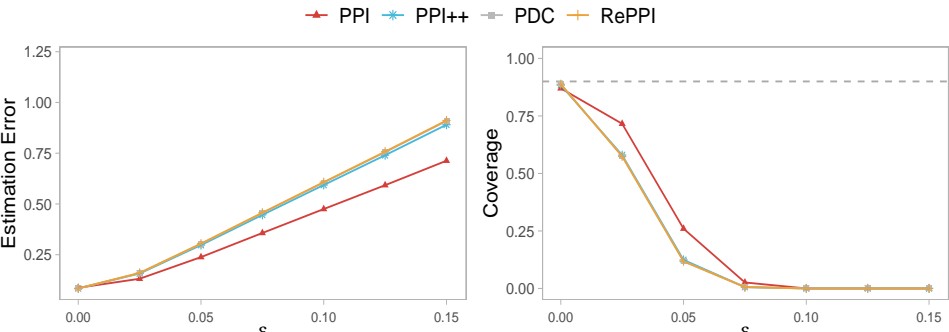

Figure 1: A running example: PPI-type methods fail under contamination.

It is important to position our work within the existing literature. There exists PPI work (Angelopoulos et al., 2023a) to address distribution shifts, particularly through domain adaptation methods that use propensity score weighting to correct for non-uniform sampling of covariates. However, these methods require the sampling probabilities to be known or well-estimated. Our work addresses a distinct but equally critical problem: outliers and contamination in the unlabeled data. The two frameworks are complementary, and a promising direction would be to combine propensity score weighting with our robust calibration to handle datasets suffering from both covariate shift and data contamination, thereby achieving a higher level of robustness.

To address these challenges, we propose a flexible and robust imputation-bias calibration method (Roica), which can be seamlessly integrated into general PPI-type SSL methods. The design of Roica is twofold, aiming to maximize the utilization of both labeled and unlabeled data. By leveraging modern robust estimation techniques and advanced prediction tools, Roica robustly extracts and calibrates information from the unlabeled data. This process ensures that the extracted information is reliable and helps improve the learning of the labeling patterns, even in the presence of outliers or corrupted values in the unlabeled data. On the other hand, to further enhance the efficiency of the learning process, Roica employs a cross-validation procedure using the labeled data for identifying the contamination proportion in the unlabeled dataset, thereby improving the overall robustness and reliability of the SSL framework. Theoretical results including the convergence and the asymptotic normality of the resulting estimator are also studied. These results are then used to guide the construction of confidence regions and the implementation of semi-supervised statistical inference.

In summary, our contributions are as follows:

- We propose Roica, a robust approach for imputation-bias calibration to the challenges posed by data contamination and distribution shifts of unlabeled data in SSL, making it a valuable tool for practical applications.

- To make a full data usage, the labeled samples are inversely used in a cross-validation procedure to determine the contamination proportion in the unlabeled ones.

- We establish the convergence and the asymptotic normality of the resulting estimator, enabling the construction of confidence regions and facilitating robust statistical inference against outliers.

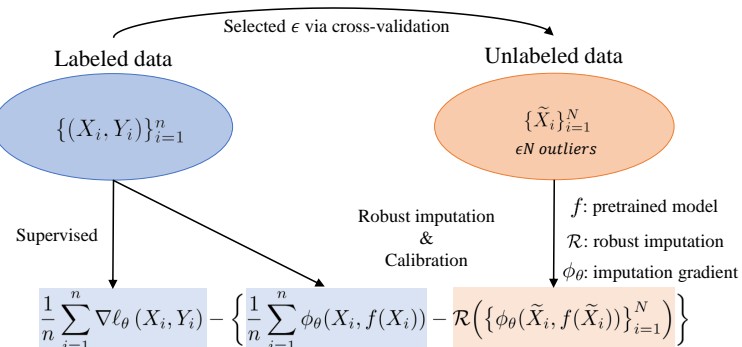

Figure 2: Illustration of the Roica procedure.

## 2 PROBLEM FORMULATION AND RELATED WORKS

Let $\mathcal{D} = \{(X_i, Y_i)\}_{i=1}^n$ be $n$ independent and identically distributed (i.i.d.) pairs drawn from the joint distribution $(X, Y) \sim P_{XY}$. Denote $P_X$ as the marginal distribution of the feature $X \in \mathbb{R}^p$ and $P_{Y|X}$ as the conditional distribution of the label $Y$ given $X$. In addition to the labeled dataset $\mathcal{D}$, an unlabeled dataset $\widetilde{\mathcal{D}} = \{\widetilde{X}_i\}_{i=1}^N$ is also available.

We are interested in the estimation and inference problems for the following M-estimation task:

$$\theta^* = \arg\min_{\theta \in \Theta} \mathbb{E}_{(X,Y) \sim P_{XY}} [\ell_\theta(X, Y)],$$

where $\Theta \subset \mathbb{R}^d$ is a convex and bounded domain, and $\ell_\theta(X, Y)$ is convex as a function of $\theta \in \Theta$. The optimal parameter vector $\theta^*$ can be regarded as the root of the gradient of the population loss, that is,

$$G(\theta^*) = \mathbb{E}_{(X,Y) \sim P_{XY}} [\nabla_\theta \ell_\theta(X, Y)] = 0. \tag{1}$$

To solve the M-estimation task, it is sufficient to provide an accurate approximation of the population gradient $G(\theta)$. Under the SSL framework, we aim to design estimation and inference methods that effectively utilize both the labeled dataset $\mathcal{D}$ and the unlabeled dataset $\widetilde{\mathcal{D}}$.

We consider the following adversarial contamination model on the unlabeled set $\widetilde{\mathcal{D}}$ in this work.

**Definition 1** (Adversarial contamination). Let $\widetilde{\mathcal{D}}_{\text{clean}} = \{\widetilde{X}_{\text{clean},i}\}_{i=1}^N$ be i.i.d. drawn from $P_X$. Someone then selectes $\epsilon$ proportion of the samples and changes them to any values. The set $\widetilde{\mathcal{D}}$ after the contamination is called a $\epsilon_*$-contamination set from distribution $P_X$. Denote $\mathcal{O}$ the set of outliers.

When the clean set $\widetilde{\mathcal{D}}_{\text{clean}}$ is available, PPI (Angelopoulos et al., 2023a) is a popular semi-supervised learning method which uses general machine-learning tools for imputation and performs valid statistical inference. The general idea of PPI (Angelopoulos et al., 2023a) is to impute the unlabeled samples by a pretrained model $f(\cdot)$, and then calibrate the imputation bias. Recent line of PPI literature develops several variants of PPI by considering more general calibration rule that minimizes the asymptotic variance of the semi-supervised estimation (Angelopoulos et al., 2023b; Gan et al., 2024; Ji et al., 2025). They can be formalized in the following form:

$$\widehat{G}^{\text{PPI}}(\theta) = \frac{1}{n} \sum_{i=1}^n \nabla \ell_\theta(X_i, Y_i) - \left\{ \frac{1}{n} \sum_{i=1}^n \phi_\theta(X_i, f(X_i)) - \frac{1}{N} \sum_{i=1}^N \phi_\theta(\widetilde{X}_i, f(\widetilde{X}_i)) \right\}, \tag{2}$$

where $\phi_\theta$ is the method-specific function called imputed gradient. Some choices of $\phi_\theta$ in the literature are summarized in Table 1.

Table 1: Different choices of $\phi_\theta$ in the PPI literature.

| Method | $\phi_\theta$ |
|---|---|
| Supervised | 0 |
| PPI (Angelopoulos et al., 2023a) | $\nabla \ell_\theta$ |
| PPI++ (Angelopoulos et al., 2023b) | $\lambda \nabla \ell_\theta$ |
| PDC (Gan et al., 2024) | $\gamma \mathbf{M} \nabla \ell_\theta$ [1] |
| RePPI (Ji et al., 2025) | $\frac{N}{n+N} \mathbf{M} \mathbb{E}\{\nabla \ell_\theta(X, Y) \mid X\}$ |

[1] The matrix $\mathbf{M}$ is selected such that it minimizes the asymptotic covariance of the resulting estimator.

# 3 METHODOLOGY: ROBUST PREDICTION-POWERED INFERENCE

To make use of the unlabeled data more safely and robustly, we propose to replace the simple average $\frac{1}{N} \sum_{i=1}^{N} \phi_\theta(\widetilde{X}_i, f(\widetilde{X}_i)))$ with a robust version,

$$\widehat{\Phi}_\theta^{\mathcal{R}} = \mathcal{R}(\{\phi_\theta(\widetilde{X}_i, f(\widetilde{X}_i))\}) \tag{3}$$

with some robust mean estimation algorithm $\mathcal{R}$ as the imputation rule. Denote the Roica gradient estimate by

$$\widehat{G}^{\mathcal{R}}(\theta) = \frac{1}{n} \sum_{i=1}^{n} \nabla \ell_\theta(X_i, Y_i) - \left\{ \frac{1}{n} \sum_{i=1}^{n} \phi_\theta(X_i, f(X_i)) - \widehat{\Phi}_\theta^{\mathcal{R}} \right\}. \tag{4}$$

When the robust estimation $\widehat{\Phi}_\theta^{\mathcal{R}}$ is close to $\Phi_\theta = \mathbb{E}_{X \sim P_X}[\phi_\theta(X_i, f(X_i))]$, the prediction-powered estimator can still achieve better estimation/inference power ignoring the shifts and outliers in the unlabeled dataset.

## 3.1 THE ROBUST MEAN IMPUTATION RULE $\mathcal{R}$

In general, the robust imputation rule $\mathcal{R}$ in Roica can be any robust mean estimator. To achieve optimal performance, we introduce the robust mean estimator, which we use in this work. In the regime of the general M-estimation, the robust mean aggregator is applied to $\{\phi_\theta(\widetilde{X}_i, f(\widetilde{X}_i))\}_{i=1}^{N}$. Denote the domain

$$\Delta_{N,\epsilon} = \left\{ w \in \mathbb{R}^N : \|w\|_1 = 1 \text{ and } 0 \leq w_i \leq \frac{1}{N(1-\epsilon)}, \, \forall i \right\}. \tag{5}$$

The key idea is finding a good weight vector $w$ in this domain so that $w_j \approx 0$ if $\widetilde{X}_j$ is poisoned and the weighted average $\Phi_{w,\theta} = \sum_{i=1}^{N} w_i \phi_\theta(\widetilde{X}_i, f(\widetilde{X}_i))$ approximates $\Phi_\theta$ well. Obviously, the feasible set ensures that the number of entries with zero weight is at most $\epsilon N$, aligning with the Huber $\epsilon$-contamination model.

The computational-efficient and strongly robust mean estimation approach Lai et al. (2016); Diakonikolas et al. (2017); Cheng et al. (2019), the *Filtering*, determines the optimal weights by minimizing the operator norm of the weighted sample covariance matrix, denoted as $\overline{\Sigma}_w(\theta)$. Specifically, the weight vector $\widehat{w}$ is defined as the solution to the optimization problem:

$$\underset{w \in \Delta_{N,\epsilon}}{\arg\min} \|\Sigma_w(\theta)\|, \tag{6}$$

where $\Sigma_w(\theta) = \sum_{i=1}^{N} w_i \phi_\theta(\widetilde{X}_i, f(\widetilde{X}_i)) \phi_\theta(\widetilde{X}_i, f(\widetilde{X}_i))^\top - \Phi_{w,\theta} \Phi_{w,\theta}^\top$ and $\epsilon$ is a hyperparameter in the algorithm which satisfies $\epsilon \geq \epsilon_*$. Under the bounded second moment assumption of the imputed graidents $\{\phi_\theta(\widetilde{X}_i, f(\widetilde{X}_i))\}$, the resulting mean estimator can achieve $O(\sqrt{\epsilon})$ bias rate.

## 3.2 FINDING THE ROOT POINT

For certain tasks like mean estimation and linear regression with certain rectifiers, there exist closed-form solutions for finding the root point of (2). We introduce them under the choice $\phi_\theta = \nabla \ell_\theta$.

- Mean estimation: $\theta^*$ is the mean of the response $Y$, i.e., $\theta^* = \mathbb{E}[Y]$. Consider the loss function $\ell_\theta(X, Y) = \|Y - \theta\|_2^2/2$, and its gradient $\nabla \ell_\theta(X, Y) = Y - \theta$. The prediction-powered estimator becomes:

$$\hat{\theta}^{\mathrm{PPI}} = \frac{1}{n}\sum_{i=1}^{n} Y_i + \frac{1}{N}\sum_{i=1}^{N} f(\widetilde{X}_i) - \frac{1}{n}\sum_{i=1}^{n} f(X_i).$$

- Linear model: consider the square loss function $\ell_\theta(X, Y) = \|Y - X^\top \theta\|_2^2/2$, and the gradient is $\nabla \ell_\theta(X, Y) = X(Y - X^\top \theta)$. Let $\mathbf{X} \in \mathbb{R}^{n \times p}$, $\tilde{\mathbf{X}} \in \mathbb{R}^{N \times p}$ be the covariate matrix of the labeled data and the unlabeled respectively, $\mathbf{Y} \in \mathbb{R}^n$ be the vector of the response of the labeled data. The prediction-powered estimate is given by:

$$\hat{\theta}^{\mathrm{PPI}} = (\tilde{\mathbf{X}}\tilde{\mathbf{X}})^{-1}\{\tilde{\mathbf{X}}^\top f(\tilde{\mathbf{X}}) - \mathbf{X}^\top(f(\mathbf{X}) - \mathbf{Y})\}.$$

For general $M$-estimation, the original PPI proposes selecting grid points in the parameter space and conducting tests at each grid point, but this approach is computationally inefficient and intractable. Angelopoulos et al. (2023b) develop efficient computational algorithms based on convex optimization. Analogously, we adopt an one-step quasi-Newton method in this work. Formally, we define our Roica estimate as:

$$\hat{\theta}^{\mathcal{R}} = \hat{\theta}_0 - \widehat{H}^{-1}\widehat{G}^{\mathcal{R}}(\hat{\theta}_0), \tag{7}$$

where $\hat{\theta}_0$ is the supervised estimator that only use labeled data and $\widehat{H}$ is the sample version of the Hessian matrix. In our simulation studies, we employ the labeled data to estimate the Hessian matrix, $\widehat{H} = \frac{1}{n}\sum_{i=1}^{n} \nabla^2 \ell_{\hat{\theta}_0}(X_i; Y_i)$.

## 3.3 SELECTION OF THE CONTAMINATION LEVEL VIA CROSS-VALIDATION

For the general $M$-estimation tasks, we can adopt cross-validation to determine the contamination level $\epsilon$ in the robust estimation algorithm $\mathcal{R}$. To make the notation clear, let $\mathcal{R}(\epsilon)$ denote the core robust estimation algorithm within Roica, parameterized by the hyperparameter $\epsilon$. The procedure is as follows:

1. Randomly partition the labeled data $\mathcal{D} = \{X_i, Y_i\}_{i=1}^{n}$ into $K$ subsets $\{\mathcal{D}_k\}_{k=1}^{K}$.

2. Let $\mathcal{E}$ be the set of candidate values for $\epsilon$. For each candidate value $\epsilon$ in $\mathcal{E}$ and for each fold $k \in [K]$, obtain the parameter estimate $\theta_{-k}^{\mathcal{R}(\epsilon)}$ by executing the Roica procedure with algorithm $\mathcal{R}(\epsilon)$ on the training data $\mathcal{D} \setminus \mathcal{D}_k$.

3. Select the contamination level estimator $\hat{\epsilon}$ by minimizing the cross-validation loss:

$$\hat{\epsilon} = \arg\min_{\epsilon \in \mathcal{E}} \sum_{k=1}^{K} \sum_{(X,Y) \in \mathcal{D}_k} \ell_{\theta_{-k}^{\mathcal{R}(\epsilon)}}(X, Y). \tag{8}$$

The cross-validation loss serves as a surrogate for the expected test loss. Therefore, the cross-validation procedure helps for the optimal choice of $\epsilon$ with the minimum expected test loss. We refer to the variant of our method that implements this cross-validation procedure as Roica-cv.

## 4 THEORETICAL RESULTS

This section provides statistical guarantees of the Roica method. For the general $M$-estimation problem, we show the asymptotic normality of $\hat{\theta}^{\mathcal{R}}$ under some mild conditions.

**Assumption 1.** (Boundness of covariance) Assume that with some positive constant $C_1 > 0$, for any $\theta \in \Theta$ such that $\|\theta - \theta^*\| \leq C/\sqrt{n}$, we have the operator norm $\|\operatorname{Cov}\{\phi_\theta(\widetilde{X}, f(\widetilde{X}))\}\|_{\mathrm{op}} \leq C_2$ for some positive constant $C_2 > 0$.

**Assumption 2.** (Regularity conditions) (a) $\theta^*$ is an interior point of $\theta$, where $\theta$ is a compact subset of $\mathbb{R}^d$. (b) $\widehat{H}$ is a consistent estimator of $H$ such that $\|\widehat{H} - H\| = o_{\mathrm{P}}(1)$. (c) $\mathbb{E}\{\|\nabla \ell_{\theta^*}(X, Y)\|^2\} < \infty$, $\mathbb{E}\{\|\phi_{\theta^*}(X, f(X))\|^2\} < \infty$. (d) $\operatorname{Cov}(\phi_{\theta^*})$ and $H$ are nonsingular. (e) There exists a neighborhood of $\theta^*$ such that $\nabla \ell_\theta(X, Y)$ is $K_1(x, y)$-Lipschitz in $\theta$ and $\phi_\theta(X, f(X))$ is $K_2(x, y)$-Lipschitz in $\theta$ with $\mathbb{E}\{K_1^2(X, Y)\} < \infty$ and $\mathbb{E}\{K_2^2(X, f(X))\} < \infty$.

We briefly discuss the role of these assumptions. Assumption 1 controls the spectral norm of the covariance matrix of the imputed gradients. This condition is fundamental to robust mean estimation (specifically the Filtering algorithm used in Roica), as it ensures that the variance of the clean data is bounded, allowing the algorithm to effectively identify outliers based on covariance inflation. Assumption 2 gathers standard regularity conditions for M-estimation (Vaart, 1998). Specifically, (a), (c), and (d) ensure the parameter space is well-behaved and the moments exist for the central limit theorem; (b) requires the Hessian estimated from labeled data to be consistent, ensuring the validity of the one-step update; and (e) imposes Lipschitz continuity to guarantee the Donsker property, which is necessary to bound the stochastic error terms in the asymptotic expansion.

Note that in the one-step Newton method, the choice of the initial parameter $\hat{\theta}_0$ is independent of the unlabeled data $\widetilde{\mathcal{D}}$. With Assumptions 1 and 2, we have the following control of the estimation error of the Filtering estimator $\mathcal{R}(\cdot)$ with hyperparameter $\epsilon = \Theta(\epsilon_* + \log(1/\tau)/N)$ (where $\Theta(\cdot)$ denotes asymptotic tight bounds). The result is summarised in the following lemma.

**Lemma 1** (Robust mean estimation error). *Under Assumptions 1 and 2, with probability at least* $1 - \tau$, *the robust mean estimation* $\widehat{\Phi}^{\mathcal{R}}_{\hat{\theta}_0}$ *satisfies that*

$$\left\|\widehat{\Phi}^{\mathcal{R}}_{\hat{\theta}_0} - \Phi_{\hat{\theta}_0}\right\|_2 \leq C\left(\sqrt{\frac{p}{N}} + \sqrt{\epsilon_*} + \sqrt{\frac{\log(1/\tau)}{N}}\right),$$

*for some positive* $C > 0$.

We present the main theorem for Roica here.

**Theorem 1.** *Assume* $n \to \infty$, $N \to \infty$, $n/N \to 0$. *And the number of outliers* $\mathcal{O}$ *satisfies that* $|\mathcal{O}| = o(N/n)$. *Denote* $H(\theta^*) = \mathbb{E}[\nabla^2 \ell_{\theta^*}(X; Y)]$ *and* $\Sigma_{\Delta,\theta^*} = \mathrm{Var}(\phi_{\theta^*}(X, f(X)) - \nabla\ell_{\theta^*}(X, Y))$ *and then*

$$\sqrt{n}(\hat{\theta}^{\mathcal{R}} - \theta^*) \xrightarrow{d} \mathcal{N}_p(0, H^{-1}(\theta^*)\Sigma_{\Delta,\theta^*}H^{-1}(\theta^*)). \tag{9}$$

Then the $(1 - \alpha)$ confidence region of $\theta^*$ is

$$\mathcal{C}_\alpha = \left\{\theta : \left\|\sqrt{n}H^{-1}(\theta)\Sigma_\theta^{-\frac{1}{2}}(\hat{\theta}^{\mathcal{R}} - \theta)\right\|^2 \leq \chi^2_{p,1-\alpha}\right\}. \tag{10}$$

Theorem 1 formally characterizes the improvement in both consistency and estimation efficiency offered by Roica. The efficiency gain is evident when comparing the asymptotic variance of our estimator to that of the supervised estimator. The asymptotic variance of the supervised estimator is $H^{-1}(\theta^*)\Sigma^2_{\sup}H^{-1}(\theta^*) = H^{-1}(\theta^*)\mathrm{Cov}(\nabla\ell_{\theta^*}(X, Y))H^{-1}(\theta^*)$, whereas the variance of the Roica estimator is determined by $H^{-1}(\theta^*)\Sigma_{\Delta,\theta^*}H^{-1}(\theta^*) = H^{-1}(\theta^*)\mathrm{Cov}(\phi_{\theta^*}(X, f(X)) - \nabla\ell_{\theta^*}(X, Y))H^{-1}(\theta^*)$. When the pre-trained model provides a good approximation of the target such that the difference term is small, we have $\Sigma^2_\Delta \prec \Sigma^2_{\sup}$. This theoretical relationship guarantees that our method not only remains consistent under contamination but also delivers more efficient estimation, yielding smaller confidence regions. This comparison underscores precisely why using only labeled data (the supervised estimator) is suboptimal, which fails to exploit the unlabeled data to achieve this statistical efficiency gain.

## 5 EXPERIMENTS

We implement the proposed Roica method in conjunction with many variants of prediction-powered inference, including PPI (Angelopoulos et al., 2023a), PPI++ (Angelopoulos et al., 2023b), PDC (Gan et al., 2024) and RePPI (Ji et al., 2025). Also, we compare these methods with the supervised estimator which only uses labeled data. Furthermore, we define two variants of Roica: Roica, an oracle variant that uses the ground-truth contamination level $\epsilon$, and Roica-cv, which estimates $\epsilon$ via cross-validation.

In the simulation studies, there are $N = 10,000$ unlabeled data points and $n = 1000$ labeled data pairs. Throughout the experiments, we fix the dimension of features as $p = 3$. For the contamination configuration, we choose $N\epsilon$ unlabeled data and add $b$ to each dimension of them, where $b$ is the shift size. The error level is $\alpha = 0.1$. We compute the estimation error by the $\ell_2$-norm loss $\|\hat{\theta} - \theta\|_2$ and calculate the volume of the confidence region. All results are based on 500 replications.

## 5.1 RESULTS FOR THE MEAN

Similar to Azriel et al. (2022), the feature $X$ is generated from the normal distribution $\mathcal{N}_p(0, \Sigma)$ with $\Sigma = (0.2^{|i-j|})_{p \times p}$ and the label is a multiple response, i.e., $Y = \mu + X^\top \beta + \rho$ where $\mu = (1, 1, 1)^\top$, $\beta = (\beta_1, \beta_2, \beta_3) \in \mathbb{R}^{p \times 3}$ and $\rho \sim \mathcal{N}_3(0_3, I_3)$ is independent of $X$. In the following experiments, we set $\beta_1 = \frac{1}{p} \cdot \mathbf{1}_p$, $\beta_2 = (0, \frac{1}{p}, \ldots, \frac{p-1}{p})^\top$ and $\beta_3 = (1, 0, \ldots, 0)^\top$.

Following the setup in Angelopoulos et al. (2023b), we use a randomized prediction function $f(X) = \mu + X^\top \beta + \eta$ with $\eta \sim \mathcal{N}_3(-0.5 \cdot \mathbf{1}_3, \sigma^2 I_3)$ with $\sigma = 0.1$ as the pre-trained model. In this setting, the target of inference is $\theta^* = \mathbb{E}[Y] = \mu$.

We compare the Roica method with the original PPI-type methods and the supervised method which refers to the simple average of labeled data. From Figure 3, it can be seen that the Roica method achieves the coverage guarantee with a smaller confidence region. In contrast, the original PPI-type methods suffer from data contamination and completely break down. In addition, the supervised method exhibits high coverage, but its confidence region is larger. For the estimation errors, the Roica method dominates the original PPI-type methods and the supervised method.

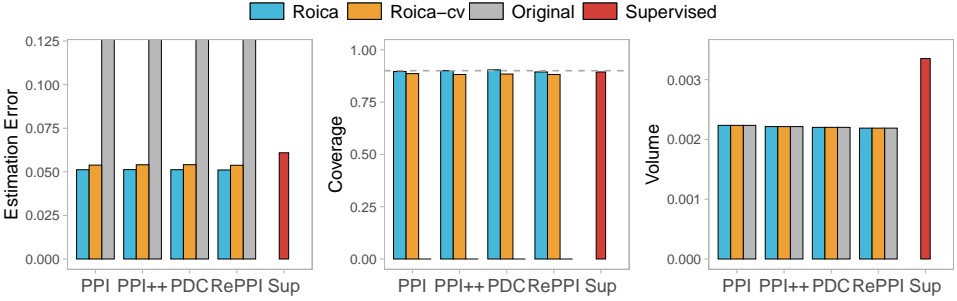

Figure 3: Empirical results of estimation error, coverage and the volume of 90% confidence regions when $b = 20$ under the contamination level $\epsilon = 0.05$ for the inference on the mean.

## 5.2 RESULTS FOR LINEAR REGRESSION

Similarly, the feature $X$ is generated from the normal distribution $\mathcal{N}_p(0, \Sigma)$ with $\Sigma = (0.2^{|i-j|})_{p \times p}$ and the response $Y = X^\top \beta_1 + (X^2)^\top \beta_2 + \rho$ where $\beta_1 = \mathbf{1}_p \in \mathbb{R}_p$, $\beta_2 = 0.3 \cdot \mathbf{1}_p \in \mathbb{R}_p$, and $\rho \sim \mathcal{N}(0, 1)$ is independent of $X$. The target of inference is defined as the regression coefficients of the least-squares solution when regressing $Y$ on $X$. In this case, $\theta^* = \beta_1$. We use a randomized predictions $f(X) = X^\top \beta_1 + (X^2)^\top \beta_2 + \eta$, with $\eta \sim \mathcal{N}(-0.5, \sigma^2)$ with $\sigma = 0.1$ to simulate the pre-trained preditions model.

Denote Roica-cv as the Roica method with the epsilon estimation via cross-validation. The candidate set of $\epsilon$ is $\mathcal{A} = \{0, 0.025, 0.05, 0.075, 0.1, 0.125, 0.15, 0.175, 0.2, 1\}$.

As shown in Figure 4, all Roica methods achieve empirical coverage statistically indistinguishable from the nominal level, confirming that the confidence regions remain valid even under data contamination. Moreover, the Roica methods produce smaller confidence regions compared to the supervised method, along with lower estimator errors and test errors. In contrast, the original PPI-type methods fail in the presence of data contamination.

## 5.3 RESULTS FOR LOGISTIC REGRESSION

The data are generated from the following logistic model, $\mathbb{P}(Y = 1 \mid X) = \frac{\exp(\beta_0 + X^\top \beta_1 + (X^2)^\top \beta_2)}{1 + \exp(\beta_0 + X^\top \beta_1 + (X^2)^\top \beta_2)}$, where $X$ follows a mixture of two multivariate normal distributions, that is, $X \sim 0.5N(\mathbf{1}_p, \Sigma) + 0.5N(-\mathbf{1}_p, \Sigma)$ with $\Sigma = (0.2^{|i-j|})_{p \times p}$. We set $(\beta_0, \beta_1^\top, \beta_2^\top)^\top = (4, 0.5 \cdot \mathbf{1}_p^\top, -\mathbf{1}_p^\top)^\top$. Consider a logistic regression working model $\mathbb{P}(Y = 1 \mid X) = \exp(\theta_0^* + $

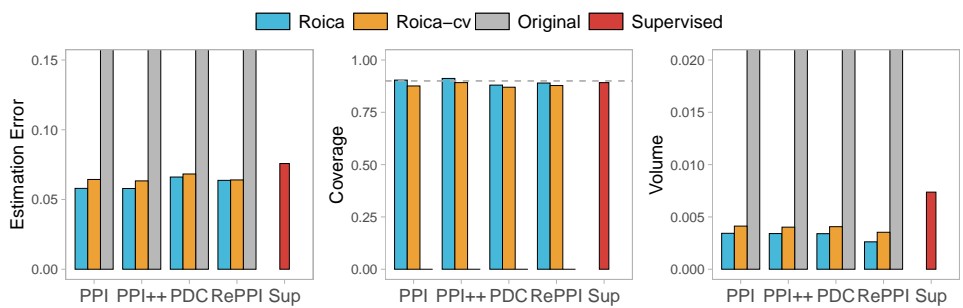

Figure 4: Empirical results of estimation error, coverage and the volume of $90\%$ confidence regions when $b = 20$ under the contamination level $\epsilon = 0.05$ for the inference on linear regression.

$X^\top \theta_1^*)/\{1 + \exp(\theta_0^* + X^\top \theta_1^*)\}$, we are interested in the estimation and inference task of the target parameter $\theta^* = (\theta_0^*, \theta_1^{*\top})^\top = \arg\min_{\theta \in \Theta} \mathbb{E}[\log\{1 + \exp(\theta_0^* + X^\top \theta_1^*)\} - Y(\theta_0^* + X^\top \theta_1^*)]$.

In line with the setting in Gan et al. (2024), we consider the prediction function $f(X) = \exp(\beta_0 + (X^2)^\top \beta_2)/(1 + \exp(\beta_0 + (X^2)^\top \beta_2))$. The empirical results of the logistic regression model are summarized in Figure 5, which showcase the robustness of the proposed method in both estimation and inference tasks.

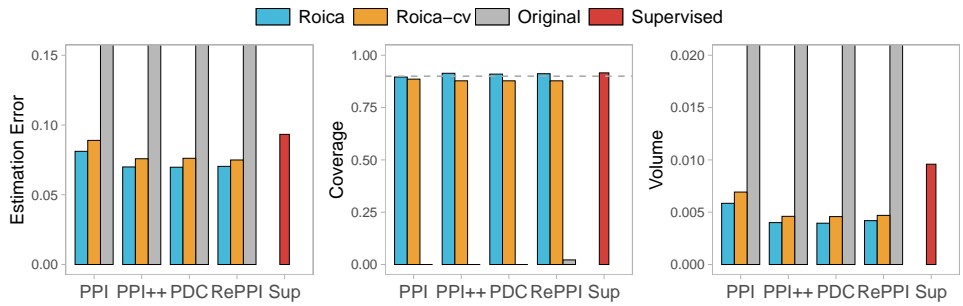

Figure 5: Empirical results of estimation error, test error, coverage and the width of $90\%$ confidence regions when $b = 20$ under the contamination level $\epsilon = 0.05$ for the inference on logistic regression.

## 6 REAL DATA ANALYSIS

### 6.1 RELATIONSHIP BETWEEN AGE, SEX, AND INCOME IN AMERICAN COMMUNITY SURVEY

We conducted an experiment using the American Community Survey Public Use Microdata Sample (PUMS) to investigate the relationship between age, sex, and income in California for the year 2019, which included a total of $380,091$ individuals. The PUMS dataset is available via the Python package *folktables* (Ding et al., 2021).

We randomly sampled $n$ data points to serve as the labeled data and treated the remaining data points $N = 380,091 - n$ as unlabeled. Following the setup in Angelopoulos et al. (2023a), the parameter of interest, $\theta^*$, is the coefficient in a linear regression of income on age and sex, defined by the population risk minimizer: $\theta^* = \arg\min_\theta \mathbb{E}[(Y - X_{\text{ols}}^\top \theta)^2]$. In this model, $Y$ denotes income (in thousands of USD), and $X_{\text{ols}}$ comprises age and sex ($X_{\text{ols}} = (X_{\text{age}}, X_{\text{sex}})$). To evaluate coverage, we took the coefficient value computed from the full dataset as the ground truth.

To obtain the predictive model $f$, we trained a XGBoost model on the PUMS dataset of 2018 with 378,817 individuals. The target Type-1 error level was $\alpha = 0.1$, and we averaged the results over 500 independent trials. For the contamination mechanism, we choose the individual whose predictive income is the highest in 2018 and randomly replace $\lfloor N\epsilon \rfloor$ individuals of 2019 with it.

The empirical results for a sample size of $n = 2000$ are summarized in Table 2. Notably, Table 2 reveals that the original PPI-type methods are highly sensitive to data contamination. In contrast, the proposed Roica method achieves valid coverage guarantees and yields more precise (smaller) confidence regions compared to the supervised benchmark.

Table 2: Empirical results of estimation error, coverage and the volume of $90\%$ confidence regions under different contamination ratios in ACS survey data.

| Method | | $\epsilon = 0$ | | | $\epsilon = 0.05$ | | | $\epsilon = 0.15$ | | |
|---|---|---|---|---|---|---|---|---|---|---|
| | | Error | Coverage | Volume | Error | Coverage | Volume | Error | Coverage | Volume |
| | Original | 0.961 | 0.922 | 0.642 | 67.649 | 0.000 | 2.338 | 168.456 | 0.000 | 11.124 |
| PPI | Roica | 0.961 | 0.922 | 0.642 | 1.004 | 0.912 | 0.642 | 1.008 | 0.910 | 0.642 |
| | Roica-cv | 1.005 | 0.922 | 0.663 | 1.048 | 0.894 | 0.667 | 1.033 | 0.896 | 0.664 |
| | Original | 0.892 | 0.908 | 0.553 | 69.043 | 0.000 | 2.400 | 221.130 | 0.000 | 18.716 |
| PPI++ | Roica | 0.892 | 0.908 | 0.553 | 0.908 | 0.910 | 0.553 | 0.910 | 0.908 | 0.553 |
| | Roica-cv | 0.932 | 0.896 | 0.583 | 0.949 | 0.892 | 0.594 | 0.953 | 0.888 | 0.592 |
| | Original | 0.897 | 0.908 | 0.549 | 42.245 | 0.000 | 1.338 | 126.858 | 0.000 | 7.277 |
| PDC | Roica | 0.897 | 0.908 | 0.549 | 0.893 | 0.916 | 0.549 | 0.891 | 0.920 | 0.549 |
| | Roica-cv | 0.956 | 0.906 | 0.581 | 0.967 | 0.898 | 0.589 | 0.958 | 0.896 | 0.588 |
| | Original | 0.896 | 0.904 | 0.549 | 42.356 | 0.000 | 1.345 | 127.188 | 0.000 | 7.321 |
| RePPI | Roica | 0.896 | 0.904 | 0.549 | 0.889 | 0.914 | 0.549 | 0.887 | 0.912 | 0.549 |
| | Roica-cv | 0.934 | 0.884 | 0.577 | 0.983 | 0.888 | 0.581 | 0.961 | 0.886 | 0.581 |
| Supervised | | 1.043 | 0.900 | 0.712 | 1.043 | 0.900 | 0.712 | 1.043 | 0.900 | 0.712 |

## 6.2 PRE-TRAINED TRANSFORMER MODEL HELPS THE ANALYSIS OF GENE EXPRESSION LEVELS

In this section, we focus on estimating the population mean to investigate how regulatory DNA influences gene expression. Specifically, we aim to estimate the population mean of expression levels driven by a population of promoters, which are regulatory DNA sequences that govern the frequency of gene transcription.

We utilize the $61,150$ experimentally validated native yeast promoters from Vaishnav et al. (2022). The sequence-to-expression transformer model developed by Vaishnav et al. (2022) is used as the prediction model, which is trained on tens of millions of random promoters to map any given sequence to an expected expression level. For each gene, we observe an 80-base-pair promoter sequence $X$ and its experimentally measured expression level $Y \in [0, 20]$. The corresponding prediction $f(X) \in [0, 20]$ is generated by the transformer model. We select all the pairs $\{(X, Y) \mid f(X) \leq 7\}$ for this study, comprising $41,651$ samples. From this population, we randomly sample $n = 500$ instances to form the labeled set and the remaining $N = 41,651 - n$ observations are treated as unlabeled. To assess coverage, we used the sample mean of the observed expression values as the ground-truth parameter. The contamination set is constructed by replacing $\lfloor N\epsilon \rfloor$ unlabeled sequences with promoter sequences randomly sampled from $\{X \mid 7 < f(X) \leq 13\}$.

Table 3 summarizes the results of all the compared methods, which confirms the effectiveness of our approach. Figure 6 shows the results of the original PPI++ and our Roica methods with varying contamination level $\epsilon$. The Roica methods achieve higher estimation precision and generate tighter confidence regions compared to the supervised benchmark.

## 7 CONCLUDING REMARKS

This paper proposes a robust prediction-powered inference (Roica) framework designed to enhance the robustness and reliability of semi-supervised learning (SSL) methods under data corruption. Traditional prediction-powered inference (PPI) fails with heterogeneous or contaminated unlabeled data. To address this, we integrate robust estimation into its imputation-calibration procedure, making it reliable even in the presence of outliers.

Through extensive numerical simulations and real-data applications, we demonstrate that Roica significantly outperforms existing PPI methods in terms of estimation accuracy, coverage, and confi-

Table 3: Empirical results of estimation error, coverage and the width of $90\%$ confidence intervals for the gene expression level estimation.

| | Method | $\epsilon = 0$ | | | $\epsilon = 0.05$ | | | $\epsilon = 0.15$ | | |
| | | Error | Coverage | Width | Error | Coverage | Volume | Error | Coverage | Width |
| --- | --- | --- | --- | --- | --- | --- | --- | --- | --- | --- |
| PPI | Original | 0.026 | 0.914 | 0.111 | 0.240 | 0.000 | 0.111 | 0.715 | 0.000 | 0.111 |
| | Roica | 0.026 | 0.914 | 0.111 | 0.027 | 0.912 | 0.111 | 0.028 | 0.914 | 0.111 |
| | Roica-cv | 0.029 | 0.884 | 0.111 | 0.032 | 0.852 | 0.111 | 0.032 | 0.850 | 0.111 |
| PPI++ | Original | 0.026 | 0.914 | 0.110 | 0.288 | 0.000 | 0.110 | 0.860 | 0.000 | 0.110 |
| | Roica | 0.026 | 0.914 | 0.110 | 0.027 | 0.918 | 0.110 | 0.028 | 0.906 | 0.110 |
| | Roica-cv | 0.028 | 0.892 | 0.110 | 0.032 | 0.850 | 0.110 | 0.032 | 0.838 | 0.110 |
| PDC | Original | 0.026 | 0.914 | 0.110 | 0.289 | 0.000 | 0.110 | 0.861 | 0.000 | 0.110 |
| | Roica | 0.026 | 0.914 | 0.110 | 0.027 | 0.918 | 0.110 | 0.028 | 0.906 | 0.110 |
| | Roica-cv | 0.028 | 0.892 | 0.110 | 0.032 | 0.856 | 0.110 | 0.032 | 0.846 | 0.110 |
| RePPI | Original | 0.026 | 0.912 | 0.110 | 0.299 | 0.000 | 0.110 | 0.891 | 0.000 | 0.110 |
| | Roica | 0.026 | 0.912 | 0.110 | 0.027 | 0.916 | 0.110 | 0.028 | 0.904 | 0.110 |
| | Roica-cv | 0.028 | 0.900 | 0.110 | 0.032 | 0.846 | 0.110 | 0.032 | 0.834 | 0.110 |
| Supervised | | 0.036 | 0.900 | 0.150 | 0.036 | 0.900 | 0.150 | 0.036 | 0.900 | 0.150 |

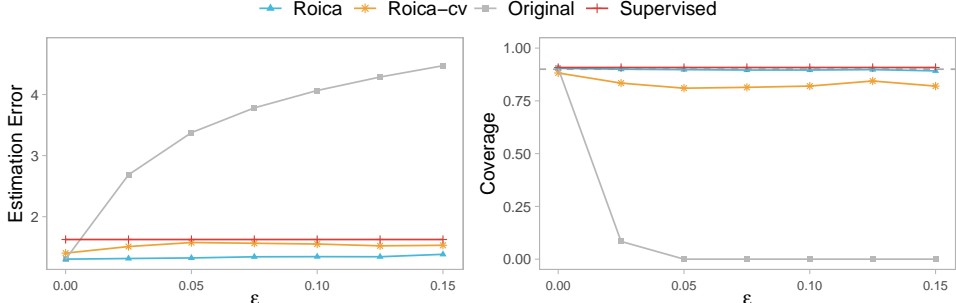

Figure 6: Empirical results of estimation error, coverage and the width of $90\%$ confidence regions with varying $\epsilon$ for the gene expression level estimation. Estimation error is log-transformed.

dence region volume in scenarios involving data contamination. Additionally, Roica exhibits robustness across various statistical models, as well as in real-world applications such as analyzing gene expression levels and survey data.

In summary, the proposed Roica framework provides a valuable tool for practical SSL applications by enhancing the robustness and reliability of prediction-powered inference under challenging data conditions. Future work may explore further extensions of Roica to other types of statistical models and investigate its performance in more complex real-world scenarios.

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

## A   SEMI-SUPERVISED INFERENCE VIA SCORE METHOD WITHOUT PARAMETRIC ESTIMATION

If we are only interested in the inference problem and a specific parametric estimation is unnecessary. We can directly apply the score-based test.

we can obtain the asymptotic normality of $\widehat{G}^{\mathcal{R}}(\theta)$.

**Theorem 2.** *Denote* $\Sigma_{\Delta,\theta} = \mathrm{Var}\left(\phi_\theta\left(X, f\left(X\right)\right) - \nabla\ell_\theta\left(X, Y\right)\right)$. *Under the conditions in Theorem 1, we have*

$$\sqrt{n}\{\widehat{G}^{\mathcal{R}}(\theta) - G(\theta)\} \xrightarrow{d} \mathcal{N}_d(0, \Sigma_{\Delta,\theta}). \tag{11}$$

*Proof.* Let $\widehat{\Delta}^f(\theta) = \frac{1}{n}\sum_{i=1}^n \left\{\phi_\theta\left(X_i, f(X_i)\right) - \nabla\ell_\theta\left(X_i, Y_i\right)\right\}$, then

$$\widehat{G}^{\mathcal{R}}(\theta) = \widehat{\Phi}^{\mathrm{R}}(\theta) - \widehat{\Delta}^f(\theta).$$

By the central limit theorem, we have

$$\sqrt{n}\{\widehat{\Delta}^f(\theta) - \Phi_\theta + G(\theta)\} \xrightarrow{d} \mathcal{N}_d(0, \Sigma_{\Delta,\theta}).$$

Note that we have $\sqrt{n}\{\widehat{\Phi}_\theta^{\mathcal{R}} - \Phi_\theta\} \xrightarrow{p} 0$. Thus

$$\sqrt{n}\widehat{G}^{\mathcal{R}}(\theta) = \sqrt{n}\{\widehat{\Phi}_\theta^{\mathcal{R}} - \Phi_\theta\} - \sqrt{n}\{\widehat{\Delta}^f(\theta) - \Phi_\theta + G(\theta)\} + \sqrt{n}G(\theta)$$
$$= o_{\mathrm{P}}(1) - \sqrt{n}\{\widehat{\Delta}^f(\theta) - G_f(\theta) + G(\theta)\} + \sqrt{n}G(\theta)$$

By Slutsky's theorem, the final result follows. $\qquad\square$

Based on Theorem 2, we can construct a $(1-\alpha)$ confidence set of $\theta^*$,

$$\mathcal{C}_\alpha = \left\{\theta : \left\|\sqrt{n}\Sigma_{\Delta,\theta}^{-\frac{1}{2}}\widehat{G}^{\mathcal{R}}(\theta)\right\|^2 \leq \chi_{d,1-\alpha}^2\right\}.$$

## B   PROOF OF THEOREM 1

*Proof.* For any measurable function $u(\cdot)$, we use the short notation

$$\mathbb{E}_n(u) = \frac{1}{n}\sum_{i=1}^n u(X_i, Y_i), \ \mathbb{G}_n(u) = \sqrt{n}\{\mathbb{E}_n(u) - \mathbb{E}(u(X,Y))\},$$

$$\widehat{\mathbb{E}}_n(u) = \frac{1}{n}\sum_{i=1}^n u(X_i, f(X_i)), \ \widehat{\mathbb{G}}_n(u) = \sqrt{n}\{\widehat{\mathbb{E}}_n(u) - \mathbb{E}(u(X,Y))\},$$

$$\widehat{\mathbb{E}}_N(u) = \frac{1}{N}\sum_{i=1}^N u(\widetilde{X}_i, f(\widetilde{X}_i)), \ \widehat{\mathbb{G}}_N(u) = \sqrt{N}\{\widehat{\mathbb{E}}_N(u) - \mathbb{E}(u(X,Y))\}.$$

According to (7), we obtain

$$\sqrt{n}\widehat{H}(\hat{\theta}^{\mathcal{R}} - \theta^*) = \sqrt{n}\left\{\widehat{H}(\hat{\theta}_0 - \theta^*) - \mathbb{E}_n(\nabla\ell_{\hat{\theta}_0}) + \widehat{\mathbb{E}}_n(\phi_{\hat{\theta}_0}) - \widehat{\Phi}_f^{\mathrm{R}}(\hat{\theta}_0)\right\}$$
$$= \sqrt{n}\widehat{H}(\hat{\theta}_0 - \theta^*) - \sqrt{n}\left\{\mathbb{E}_n(\nabla\ell_{\hat{\theta}_0} - \nabla\ell_{\theta^*}) - \widehat{\mathbb{E}}_n(\phi_{\hat{\theta}_0} - \phi_{\theta^*}) - \widehat{\mathbb{E}}_N(\phi_{\hat{\theta}_0} - \phi_{\theta^*})\right\}$$
$$+ \sqrt{n}\left\{\widehat{\mathbb{E}}_n(\phi_{\theta^*}) - \mathbb{E}_n(\nabla\ell_{\theta^*}) + \widehat{\mathbb{E}}_N(\phi_{\theta^*}) - \widehat{\mathbb{E}}_N(\phi_{\hat{\theta}_0}) + \widehat{\Phi}_f^{\mathrm{R}}(\hat{\theta}_0)\right\}$$
$$:= \mathrm{I} - \mathrm{II} + \mathrm{III}.$$

By Assumption 2(e) and Example 19.7 in Vaart (1998), we know function class $\{\nabla\ell_\theta(\cdot,\cdot) : \theta \in \Theta\}$ is a $P_{X,Y}$-Donsker class. Next, Lemma 19.24 of Vaart (1998) implies

$$\mathbb{G}_n(\nabla\ell_{\hat{\theta}_0} - \nabla\ell_{\theta^*}) = o_{\mathrm{P}}(1) \tag{12}$$

Under Assumption 2, the supervised estimator $\hat{\theta}_0$ is $O_{\mathrm{P}}(1/\sqrt{n})$ by the theory of supervised $M$-estimation, see Vaart (1998). Hence, by Assumption 2 (b) and Taylor expansion, we have

$$\sqrt{n}\mathbb{E}_n(\nabla\ell_{\hat{\theta}_0} - \nabla\ell_{\theta^*}) = \sqrt{n}\mathbb{E}(\nabla\ell_{\hat{\theta}_0} - \nabla\ell_{\theta^*}) + o_{\mathrm{P}}(1)$$
$$= \sqrt{n}H(\hat{\theta}_0 - \theta^*) + \sqrt{n}o_{\mathrm{P}}(\|\hat{\theta}_0 - \theta^*\|) + o_{\mathrm{P}}(1) \tag{13}$$
$$= \sqrt{n}\widehat{H}(\hat{\theta}_0 - \theta^*) + o_{\mathrm{P}}(1).$$

Similar as (12), we derive

$$\widehat{\mathbb{G}}_n(\phi_{\hat{\theta}_0} - \phi_{\theta^*}) = o_{\mathrm{P}}(1), \ \widehat{\mathbb{G}}_N(\phi_{\hat{\theta}_0} - \phi_{\theta^*}) = o_{\mathrm{P}}(1),$$

thus we obtain

$$\widehat{\mathbb{E}}_n(\phi_{\hat{\theta}_0} - \phi_{\theta^*}) = \mathbb{E}_n(\phi_{\hat{\theta}_0} - \phi_{\theta^*}) + o_{\mathrm{P}}(1/\sqrt{n}), \tag{14}$$

$$\widehat{\mathbb{E}}_N(\phi_{\hat{\theta}_0} - \phi_{\theta^*}) = \mathbb{E}_N(\phi_{\hat{\theta}_0} - \phi_{\theta^*}) + o_{\mathrm{P}}(1/\sqrt{N}). \tag{15}$$

Combining (13), (14) and (15), as $n/N = o_{\mathrm{P}}(1)$,

$$\mathrm{I} - \mathrm{II} = o_{\mathrm{P}}(1). \tag{16}$$

For the term III, by the central limited thereom,

$$\sqrt{n}\left\{\widehat{\mathbb{E}}_n(\phi_{\theta^*}) - \mathbb{E}_n(\nabla\ell_{\theta^*}) - \mathbb{E}(\phi_{\theta^*}) + \mathbb{E}(\nabla\ell_{\theta^*})\right\} \xrightarrow{d} \mathcal{N}_p(0, \Sigma_{\Delta,\theta^*}), \quad (17)$$

where $\Sigma_{\Delta,\theta^*} = \text{Var}(\phi_{\theta^*}(X, f(X)) - \nabla\ell_{\theta^*}(X, Y))$.

Note that we have

$$\widehat{\Phi}_f^{\text{R}}(\hat{\theta}_0) - \widehat{\mathbb{E}}_N(\phi_{\hat{\theta}_0}) = o_{\text{P}}(1/\sqrt{n}). \quad (18)$$

Also, by the central limit theorem, we can prove $\widehat{\mathbb{E}}_N(\phi_{\theta^*}) = O_{\text{P}}(1/\sqrt{N})$. Thus , (17), (18) together implies

$$\text{III} \xrightarrow{d} \mathcal{N}_p(0, \Sigma_{\Delta,\theta^*}). \quad (19)$$

Combining (16) and (19), by Slutsky's theorem, we conclude

$$\sqrt{n}\widehat{H}(\hat{\theta}^{\mathcal{R}} - \theta^*) \xrightarrow{d} \mathcal{N}_p(0, \Sigma_{\Delta,\theta^*}).$$

Finally, by Slutsky's theorem and Assumption (2)(b), we complete the proof. □

## C  ADDITIONAL NUMERICAL RESULTS

### C.1  SUPPLEMENTARY RESULTS FOR SYNTHETIC DATA

Based on the experimental setup established in Section 5, this section investigates additional synthetic data scenarios to further validate our methodology.

Figure 7, 8, 9 exhibit the performance of our proposed Roica method compared with other methods when $\epsilon = 0,\ 0.15$. The estimation error is transformed by $\log(\log(100x))$ and the volume is transformed by $\log(\log(2000x))$. As can be seen, almost all Roica methods can construct smaller confidence regions with the coverage guarantee and give more precise estimators, while the original methods fail.

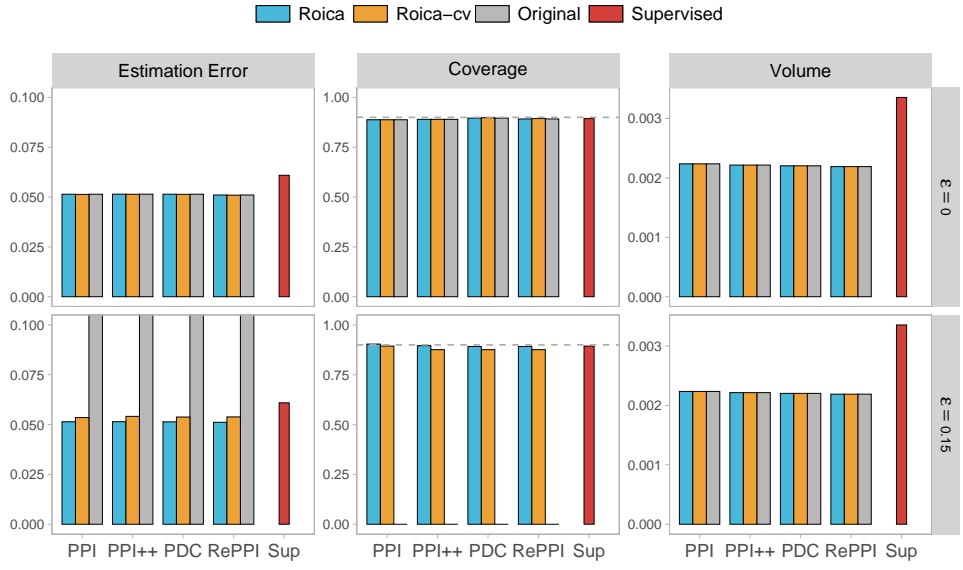

Figure 7: Empirical results of estimation error, coverage and the volume of $90\%$ confidence regions when $b = 20$ under the contamination level $\epsilon = 0,\ 0.15$ for the inference on mean.

### C.2  OTHER CHOICES FOR PREDICTOR WHEN PRE-TRAINED MODEL IS NOT AVAILABLE

In fact, it is quite important to make prediction-powered inference when a pre-trained model is not available. There are two strategies we adopted here. The first one is inspired by Zrnic & Candès

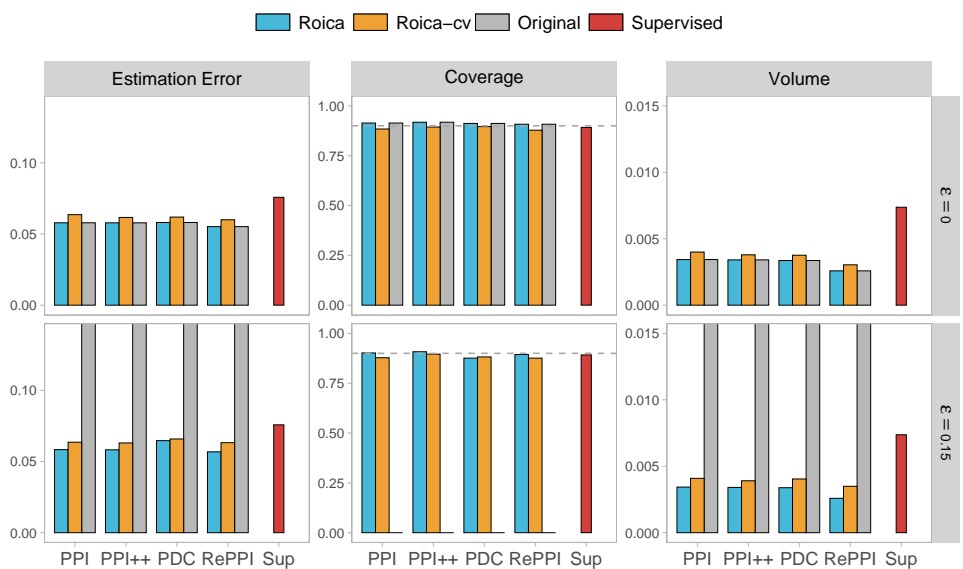

Figure 8: Empirical results of estimation error, test error, coverage and the volume of $90\%$ confidence regions when $b = 20$ under the contamination level $\epsilon = 0,\ 0.15$ for the inference on linear regression.

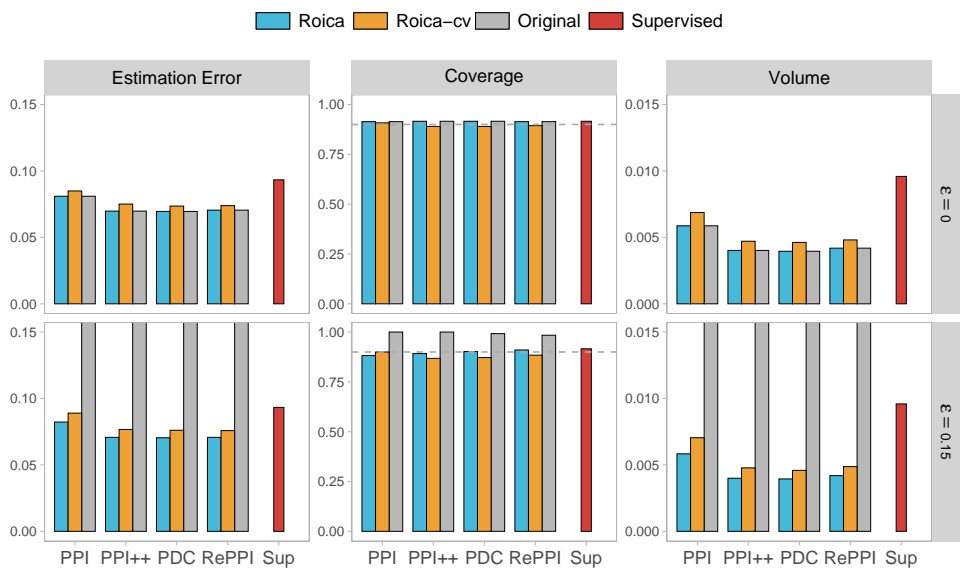

Figure 9: Empirical results of estimation error, test error, coverage and the volume of $90\%$ confidence regions when $b = 20$ under the contamination level $\epsilon = 0,\ 0.15$ for the inference on logistic regression.

(2024), we adopt the cross-fitting strategy to fit a model $f$ using the labeled data and the second one is splitting the labeled data into two equal subsets and fitting a model on one of them. We conduct more experiments on two strategies. For linear and logistic regression, we train gradient-boosted trees via XGBoost to obtain a model $f$ and the least squares method is adopted for mean estimation. All settings are the same as Section 5.

From Table 4, we can see that our Roica still works and remains stable when using the cross-fitting prediction strategy instead of a pre-trained model. In the scenario of Table 5 and Table 6, the coverage of our Roica is less than the target level, but is robust than the original PPI-type methods.

Table 4: Empirical results of estimation error, coverage and the volume of $90\%$ confidence regions under different contamination ratios when $b = 20$ for the inference of mean with the cross-fitting strategy.

| Method | | $\epsilon = 0$ | | | $\epsilon = 0.05$ | | | $\epsilon = 0.15$ | | |
|---|---|---|---|---|---|---|---|---|---|---|
| | | Error | Coverage | Volume | Error | Coverage | Volume | Error | Coverage | Volume |
| PPI | Original | 0.051 | 0.896 | 0.222 | 1.498 | 0.000 | 0.222 | 4.491 | 0.000 | 0.222 |
| | Roica | 0.051 | 0.896 | 0.222 | 0.051 | 0.898 | 0.222 | 0.051 | 0.892 | 0.222 |
| | Roica-cv | 0.051 | 0.894 | 0.222 | 0.054 | 0.878 | 0.222 | 0.054 | 0.878 | 0.222 |
| PPI++ | Original | 0.051 | 0.894 | 0.221 | 1.345 | 0.000 | 0.221 | 4.032 | 0.000 | 0.221 |
| | Roica | 0.051 | 0.894 | 0.221 | 0.051 | 0.902 | 0.221 | 0.051 | 0.896 | 0.221 |
| | Roica-cv | 0.051 | 0.896 | 0.221 | 0.054 | 0.882 | 0.221 | 0.054 | 0.882 | 0.221 |
| PDC | Original | 0.051 | 0.898 | 0.220 | 1.356 | 0.000 | 0.220 | 4.066 | 0.000 | 0.220 |
| | Roica | 0.051 | 0.898 | 0.220 | 0.051 | 0.900 | 0.220 | 0.051 | 0.898 | 0.220 |
| | Roica-cv | 0.051 | 0.900 | 0.220 | 0.054 | 0.882 | 0.220 | 0.054 | 0.886 | 0.220 |
| RePPI | Original | 0.051 | 0.894 | 0.215 | 1.499 | 0.000 | 0.215 | 4.496 | 0.000 | 0.215 |
| | Roica | 0.051 | 0.894 | 0.215 | 0.051 | 0.894 | 0.215 | 0.051 | 0.884 | 0.215 |
| | Roica-cv | 0.051 | 0.894 | 0.215 | 0.054 | 0.870 | 0.215 | 0.054 | 0.872 | 0.215 |
| Supervised | | 0.061 | 0.894 | 0.335 | 0.061 | 0.894 | 0.335 | 0.061 | 0.894 | 0.335 |

Table 5: Empirical results of estimation error, coverage and the volume of $90\%$ confidence regions under different contamination ratios when $b = 20$ for linear regression with the cross-fitting strategy.

| Method | | $\epsilon = 0$ | | | $\epsilon = 0.05$ | | | $\epsilon = 0.15$ | | |
|---|---|---|---|---|---|---|---|---|---|---|
| | | Error | Coverage | Volume | Error | Coverage | Volume | Error | Coverage | Volume |
| PPI | Original | 0.070 | 0.882 | 0.005 | 1.432 | 0.000 | 0.030 | 1.450 | 0.000 | 0.031 |
| | Roica | 0.070 | 0.882 | 0.005 | 0.073 | 0.850 | 0.005 | 0.073 | 0.864 | 0.005 |
| | Roica-cv | 0.073 | 0.878 | 0.006 | 0.074 | 0.876 | 0.006 | 0.075 | 0.868 | 0.006 |
| PPI++ | Original | 0.066 | 0.886 | 0.005 | 62.238 | 0.000 | $\geq 100$ | $\geq 100$ | 0.000 | $\geq 100$ |
| | Roica | 0.066 | 0.886 | 0.005 | 0.068 | 0.862 | 0.005 | 0.068 | 0.882 | 0.005 |
| | Roica-cv | 0.069 | 0.878 | 0.005 | 0.071 | 0.872 | 0.005 | 0.071 | 0.874 | 0.005 |
| PDC | Original | 0.066 | 0.882 | 0.004 | 48.767 | 0.000 | $\geq 100$ | $\geq 100$ | 0.000 | $\geq 100$ |
| | Roica | 0.066 | 0.882 | 0.004 | 0.100 | 0.800 | 0.005 | 0.095 | 0.814 | 0.005 |
| | Roica-cv | 0.069 | 0.862 | 0.005 | 0.084 | 0.848 | 0.006 | 0.079 | 0.858 | 0.006 |
| RePPI | Original | 0.066 | 0.884 | 0.004 | 52.988 | 0.000 | $\geq 100$ | $\geq 100$ | 0.000 | $\geq 100$ |
| | Roica | 0.066 | 0.884 | 0.004 | 0.096 | 0.788 | 0.005 | 0.121 | 0.794 | 0.006 |
| | Roica-cv | 0.070 | 0.880 | 0.005 | 0.082 | 0.842 | 0.006 | 0.088 | 0.868 | 0.006 |
| Supervised | | 0.076 | 0.892 | 0.007 | 0.076 | 0.892 | 0.007 | 0.076 | 0.892 | 0.007 |

## C.3 RESULTS FOR REAL DATA

### C.3.1 ADDITIONAL RESULTS ON ACS SURVEY DATA

In this section, we consider the same setup in 6.1 and run more experiments under different settings. We also discuss the performance of our proposed Roica with different sample sizes of labeled data $n$ in Table 10 and Table 11. All results can confirm the robustness and effectiveness of our methods.

### C.3.2 ADDITIONAL RESULTS ON THE MEAN OF GENE-EXPRESSION LEVELS

Supplementary Results of aggressive attack In this subsection, we conduct more experiments on the dataset of gene-expression levels. Figure 11 shows how other original PPI-typed methods and our Roica methods perform when the contamination $\epsilon$ varies. It is obvious that our Roica methods are robust against contamination and construct smaller confidence intervals. We also analyze the

Table 6: Empirical results of estimation error, coverage and the volume of $90\%$ confidence regions under different contamination ratios when $b = 20$ for logistic regression with the cross-fitting strategy.

| | Method | $\epsilon = 0$ | | | $\epsilon = 0.05$ | | | $\epsilon = 0.15$ | | |
| --- | --- | --- | --- | --- | --- | --- | --- | --- | --- | --- |
| | | Error | Coverage | Volume | Error | Coverage | Volume | Error | Coverage | Volume |
| PPI | Original | 0.079 | 0.912 | 0.006 | 1.832 | 0.000 | 1.384 | 5.491 | 1.000 | $\geq 100$ |
| | Roica | 0.079 | 0.912 | 0.006 | 0.105 | 0.186 | 0.005 | 0.154 | 0.000 | 0.005 |
| | Roica-cv | 0.090 | 0.842 | 0.007 | 0.097 | 0.700 | 0.008 | 0.094 | 0.902 | 0.010 |
| PPI++ | Original | 0.069 | 0.894 | 0.004 | 1.370 | 0.000 | 0.266 | 4.222 | 1.000 | $\geq 100$ |
| | Roica | 0.069 | 0.894 | 0.004 | 0.080 | 0.536 | 0.004 | 0.105 | 0.082 | 0.004 |
| | Roica-cv | 0.074 | 0.848 | 0.005 | 0.086 | 0.648 | 0.006 | 0.094 | 0.832 | 0.009 |
| PDC | Original | 0.069 | 0.896 | 0.004 | 1.282 | 0.000 | 0.186 | 3.846 | 1.000 | 84.386 |
| | Roica | 0.069 | 0.896 | 0.004 | 0.077 | 0.642 | 0.004 | 0.091 | 0.408 | 0.004 |
| | Roica-cv | 0.075 | 0.840 | 0.004 | 0.084 | 0.726 | 0.006 | 0.091 | 0.746 | 0.007 |
| RePPI | Original | 0.069 | 0.902 | 0.004 | 0.768 | 0.004 | 0.035 | 2.306 | 0.170 | 9.889 |
| | Roica | 0.069 | 0.902 | 0.004 | 0.077 | 0.662 | 0.004 | 0.097 | 0.264 | 0.004 |
| | Roica-cv | 0.077 | 0.838 | 0.005 | 0.085 | 0.718 | 0.006 | 0.094 | 0.738 | 0.008 |
| Supervised | | 0.093 | 0.916 | 0.010 | 0.093 | 0.916 | 0.010 | 0.093 | 0.916 | 0.010 |

Table 7: Empirical results of estimation error, coverage and the volume of $90\%$ confidence regions under different contamination ratios when $b = 20$ for the inference of mean with the splitting strategy.

| | Method | $\epsilon = 0$ | | | $\epsilon = 0.05$ | | | $\epsilon = 0.15$ | | |
| --- | --- | --- | --- | --- | --- | --- | --- | --- | --- | --- |
| | | Error | Coverage | Volume | Error | Coverage | Volume | Error | Coverage | Volume |
| PPI | Original | 0.094 | 0.890 | 0.012 | 0.173 | 0.388 | 0.012 | 0.440 | 0.046 | 0.012 |
| | Roica | 0.094 | 0.890 | 0.012 | 0.094 | 0.890 | 0.012 | 0.094 | 0.888 | 0.012 |
| | Roica-cv | 0.094 | 0.890 | 0.012 | 0.094 | 0.878 | 0.012 | 0.094 | 0.894 | 0.012 |
| PPI++ | Original | 0.091 | 0.900 | 0.011 | 0.380 | 0.238 | 0.011 | 1.094 | 0.090 | 0.011 |
| | Roica | 0.091 | 0.900 | 0.011 | 0.091 | 0.900 | 0.011 | 0.091 | 0.898 | 0.011 |
| | Roica-cv | 0.091 | 0.900 | 0.011 | 0.089 | 0.912 | 0.011 | 0.091 | 0.886 | 0.011 |
| PDC | Original | 0.072 | 0.896 | 0.006 | 1.648 | 0.000 | 0.006 | 4.947 | 0.000 | 0.006 |
| | Roica | 0.072 | 0.896 | 0.006 | 0.072 | 0.888 | 0.006 | 0.072 | 0.886 | 0.006 |
| | Roica-cv | 0.072 | 0.894 | 0.006 | 0.076 | 0.860 | 0.006 | 0.076 | 0.858 | 0.006 |
| RePPI | Original | 0.072 | 0.890 | 0.006 | 1.731 | 0.000 | 0.006 | 5.194 | 0.000 | 0.006 |
| | Roica | 0.072 | 0.890 | 0.006 | 0.072 | 0.892 | 0.006 | 0.072 | 0.886 | 0.006 |
| | Roica-cv | 0.072 | 0.890 | 0.006 | 0.075 | 0.864 | 0.006 | 0.076 | 0.852 | 0.006 |
| Supervised | | 0.066 | 0.894 | 0.004 | 0.066 | 0.894 | 0.004 | 0.066 | 0.894 | 0.004 |

performance of our proposed Roica methods with different $n$. Figure 12 and Figure 13 report the empirical results of all methods when the contamination level $\epsilon = 0.05, 0.15$.

We also consider the more aggressive attack on the gene-expression level task. In detail, we construct the contamination set by replacing $\lfloor N\epsilon \rfloor$ unlabeled sequences with promoter sequences randomly sampled from $\{X \mid f(X) > 13\}$. The results are summarized in Table 12 and Figure 14. Under the aggressive attack, the original PPI-type methods break down when there exists contamination.

### C.4 RESULTS FOR HIGHER CONTAMINATION RATIOS

Similar the setup in Section 5, we conduct more experiments on linear regression with higher contamination ratios. As shown in Table C.4, these results confirm that our method performs well across a wide range of noise levels, maintaining robust performance even at $\epsilon = 0.4$.

### C.5 RESULTS FOR THE ADVERSARIAL ATTACK

Inspired by the work of Li et al. (2022) and Diakonikolas & Kane (2021), we introduce a more challenging adversarial contamination scheme aimed at corrupting the fundamental structure of the covariate space $(X)$. The specific methodology is as follows: randomly selected samples are per-

Table 8: Empirical results of estimation error, coverage and the volume of $90\%$ confidence regions under different contamination ratios when $b = 20$ for linear regression with the splitting strategy.

| Method | | $\epsilon = 0$ | | | $\epsilon = 0.05$ | | | $\epsilon = 0.15$ | | |
| | | Error | Coverage | Volume | Error | Coverage | Volume | Error | Coverage | Volume |
|---|---|---|---|---|---|---|---|---|---|---|
| PPI | Original | 0.101 | 0.882 | 0.015 | 1.449 | 0.000 | 0.092 | 1.468 | 0.000 | 0.094 |
| | Roica | 0.101 | 0.882 | 0.015 | 0.102 | 0.880 | 0.015 | 0.102 | 0.876 | 0.015 |
| | Roica-cv | 0.093 | 0.890 | 0.013 | 0.092 | 0.884 | 0.012 | 0.679 | 0.890 | $\geq 100$ |
| PPI++ | Original | 0.095 | 0.866 | 0.013 | 66.077 | 0.000 | $\geq 100$ | $\geq 100$ | 0.000 | $\geq 100$ |
| | Roica | 0.095 | 0.866 | 0.013 | 0.095 | 0.868 | 0.013 | 0.095 | 0.868 | 0.013 |
| | Roica-cv | 0.092 | 0.866 | 0.012 | 0.093 | 0.870 | 0.012 | 0.932 | 0.876 | $\geq 100$ |
| PDC | Original | 0.095 | 0.854 | 0.013 | 48.058 | 0.000 | $\geq 100$ | $\geq 100$ | 0.000 | $\geq 100$ |
| | Roica | 0.095 | 0.854 | 0.013 | 0.101 | 0.832 | 0.013 | 0.103 | 0.836 | 0.013 |
| | Roica-cv | 0.093 | 0.866 | 0.012 | 0.095 | 0.844 | 0.011 | 0.097 | 0.856 | 0.012 |
| RePPI | Original | 0.097 | 0.856 | 0.012 | 51.070 | 0.000 | $\geq 100$ | $\geq 100$ | 0.000 | $\geq 100$ |
| | Roica | 0.097 | 0.856 | 0.012 | 0.104 | 0.820 | 0.012 | 0.106 | 0.822 | 0.013 |
| | Roica-cv | 0.094 | 0.858 | 0.011 | 0.189 | 0.838 | 2.497 | 0.097 | 0.858 | 0.011 |
| Supervised | | 0.076 | 0.892 | 0.007 | 0.076 | 0.892 | 0.007 | 0.076 | 0.892 | 0.007 |

Table 9: Empirical results of estimation error, coverage and the volume of $90\%$ confidence regions under different contamination ratios when $b = 20$ for logistic regression with the splitting strategy.

| Method | | $\epsilon = 0$ | | | $\epsilon = 0.05$ | | | $\epsilon = 0.15$ | | |
| | | Error | Coverage | Volume | Error | Coverage | Volume | Error | Coverage | Volume |
|---|---|---|---|---|---|---|---|---|---|---|
| PPI | Original | 0.114 | 0.920 | 0.016 | 1.848 | 0.000 | 3.835 | 5.536 | 1.000 | $\geq 100$ |
| | Roica | 0.114 | 0.920 | 0.016 | 0.114 | 0.914 | 0.015 | 0.115 | 0.908 | 0.015 |
| | Roica-cv | 0.111 | 0.900 | 0.014 | 0.110 | 0.902 | 0.014 | 0.111 | 0.894 | 0.014 |
| PPI++ | Original | 0.103 | 0.920 | 0.011 | 1.434 | 0.000 | 0.910 | 4.415 | 1.000 | $\geq 100$ |
| | Roica | 0.103 | 0.920 | 0.011 | 0.103 | 0.908 | 0.011 | 0.103 | 0.906 | 0.011 |
| | Roica-cv | 0.104 | 0.886 | 0.011 | 0.105 | 0.888 | 0.011 | 0.104 | 0.880 | 0.011 |
| PDC | Original | 0.102 | 0.906 | 0.011 | 1.343 | 0.000 | 0.623 | 4.021 | 1.000 | $\geq 100$ |
| | Roica | 0.102 | 0.906 | 0.011 | 0.102 | 0.912 | 0.011 | 0.103 | 0.890 | 0.011 |
| | Roica-cv | 0.105 | 0.874 | 0.011 | 0.104 | 0.884 | 0.011 | 0.104 | 0.858 | 0.011 |
| RePPI | Original | 0.104 | 0.912 | 0.011 | 0.793 | 0.046 | 0.469 | 2.321 | 0.410 | $\geq 100$ |
| | Roica | 0.104 | 0.912 | 0.011 | 0.106 | 0.854 | 0.011 | 0.114 | 0.768 | 0.012 |
| | Roica-cv | 0.105 | 0.894 | 0.011 | 0.102 | 0.848 | 0.011 | 0.102 | 0.826 | 0.011 |
| Supervised | | 0.093 | 0.916 | 0.010 | 0.093 | 0.916 | 0.010 | 0.093 | 0.916 | 0.010 |

turbed along the direction of the dataset's first principal component, formulated as $\tilde{X}_i^o = \tilde{X}_i + \delta v_1$ where $v_1$ represents the first principal component direction of $\{\tilde{X}_i\}_{i=1}^N$, and $\delta$ denotes the perturbation intensity. The results for linear regression presented in Table C.5 conclusively demonstrate that our method (Roica) maintains robust performance even under this stringent adversarial setting, outperforming the comparison methods.

## C.6 ANALYSIS OF THE SENSITIVITY OF THE PREDICTION MODEL

To quantitatively assess the sensitivity to variations in the prediction model, we have conducted a dedicated analysis in the linear regression setting. We systematically varied the parameters of the predictive model $f(X) = X^\top \beta_1 + (X^2)^\top \beta_2 + \eta, \quad \eta \sim \mathcal{N}(-\mu_\eta, \sigma_\eta)$.

The results presented in Figure 15 illustrate how the performance of our method changes with the mean $\mu_\eta$ (capturing systematic bias) and the variance $\sigma_\eta$ (capturing random inaccuracy) of the predictor. Indeed, as theoretically anticipated, the performance of all methods is influenced by increasing model bias($\mu_\eta$) and variance ($\sigma_\eta^2$). A consistent observation is that Roica (and Roica-cv) robustly maintains valid coverage across all perturbation levels. Concurrently, as the predictor's inaccuracy grows whether through systematic bias or random noise, the estimation error and the volume of the confidence region for our method gradually decrease. This phenomenon is intuitive: as the model predictions become less reliable, the information they provide becomes inherently less certain.

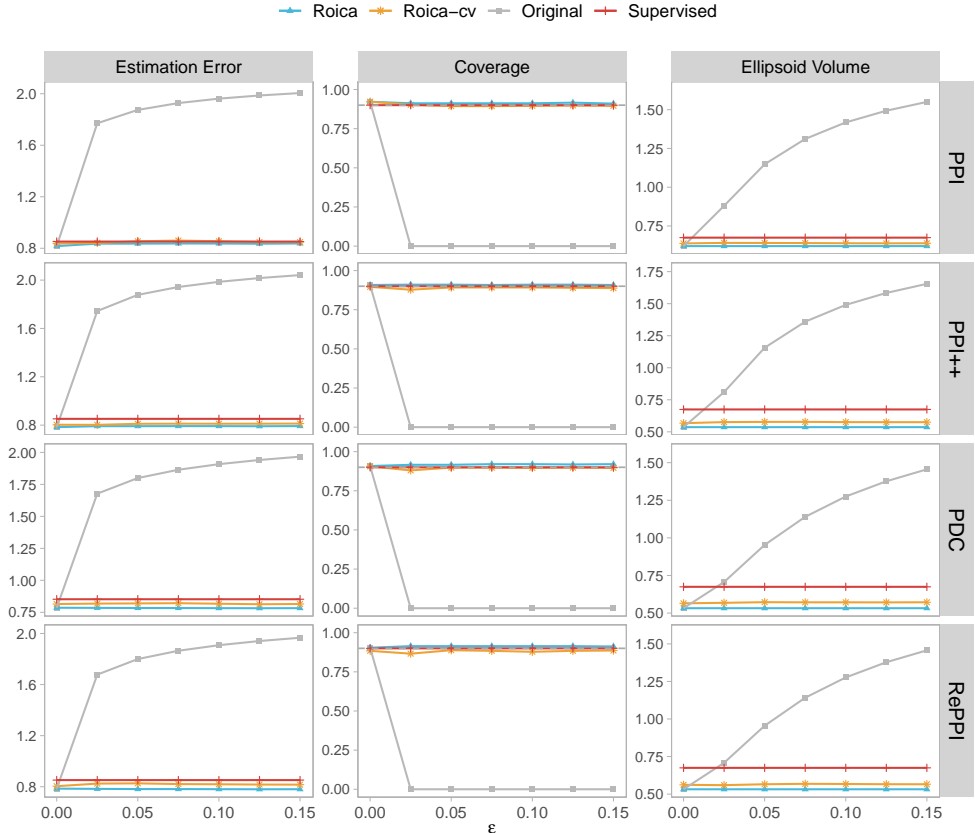

Figure 10: Empirical results of estimation error, coverage and the volume of $90\%$ confidence regions with varying $\epsilon$ when $n = 2000$ in ACS survey data.

Table 10: Empirical results of estimation error, coverage and the volume of $90\%$ confidence regions under contamination level $\epsilon = 0.15$ for $n = 1000, 3000, 4000$ in ACS survey data.

| | Method | | $n = 1000$ | | | $n = 3000$ | | | $n = 4000$ | |
|---|---|---|---|---|---|---|---|---|---|---|
| | | Erorr | Coverage | Volume | Eroor | Coverage | Volume | Erorr | Coverage | Volume |
| PPI | Original | 168.531 | 0.000 | 22.191 | 168.480 | 0.000 | 7.404 | 168.474 | 0.000 | 5.551 |
| | Roica | 1.565 | 0.920 | 1.268 | 0.856 | 0.882 | 0.430 | 0.757 | 0.914 | 0.325 |
| | Roica-cv | 1.563 | 0.896 | 1.309 | 0.876 | 0.878 | 0.447 | 0.765 | 0.886 | 0.337 |
| PPI++ | Original | 221.956 | 0.000 | 37.531 | 220.821 | 0.000 | 12.415 | 220.291 | 0.000 | 9.263 |
| | Roica | 1.405 | 0.900 | 1.092 | 0.773 | 0.882 | 0.371 | 0.686 | 0.908 | 0.280 |
| | Roica-cv | 1.438 | 0.884 | 1.170 | 0.814 | 0.872 | 0.397 | 0.717 | 0.886 | 0.301 |
| PDC | Original | 129.318 | 0.000 | 15.260 | 126.411 | 0.000 | 4.784 | 125.891 | 0.000 | 3.546 |
| | Roica | 1.388 | 0.910 | 1.079 | 0.754 | 0.880 | 0.369 | 0.670 | 0.914 | 0.279 |
| | Roica-cv | 1.435 | 0.892 | 1.150 | 0.807 | 0.864 | 0.396 | 0.711 | 0.878 | 0.299 |
| RePPI | Original | 129.661 | 0.000 | 15.428 | 127.000 | 0.000 | 4.824 | 126.766 | 0.000 | 3.589 |
| | Roica | 1.378 | 0.912 | 1.078 | 0.758 | 0.878 | 0.369 | 0.673 | 0.910 | 0.279 |
| | Roica-cv | 1.405 | 0.890 | 1.150 | 0.805 | 0.862 | 0.396 | 0.697 | 0.880 | 0.296 |
| Supervised | | 1.544 | 0.894 | 1.413 | 0.868 | 0.900 | 0.477 | 0.757 | 0.912 | 0.360 |

# D    IMPLEMENTATION DETAILS OF ROICA

## D.1    ROBUST ESIMATION PROCEDURE

This section details the robust Filtering algorithm used to compute the weights $w$ in Equation 6 of the main text. We exhibit the key step of Filtering procedure(Zhu et al., 2023) in Algorithm 1.

Table 11: Empirical results of estimation error, coverage and the volume of $90\%$ confidence regions under contamination level $\epsilon = 0.15$ for $n = 5000, 6000, 7000$ in ACS survey data.

| | | $n = 5000$ | | | $n = 6000$ | | | $n = 7000$ | | |
| --- | --- | --- | --- | --- | --- | --- | --- | --- | --- | --- |
| | Method | Est.err | Coverage | Volume | Est.err | Coverage | Volume | Est.err | Coverage | Volume |
| PPI | Original | 168.501 | 0.000 | 4.440 | 168.506 | 0.000 | 3.700 | 168.497 | 0.000 | 3.173 |
| | Roica | 0.664 | 0.904 | 0.260 | 0.608 | 0.916 | 0.217 | 0.552 | 0.920 | 0.187 |
| | Roica-cv | 0.663 | 0.878 | 0.268 | 0.594 | 0.900 | 0.224 | 0.572 | 0.898 | 0.193 |
| PPI++ | Original | 219.714 | 0.000 | 7.373 | 219.152 | 0.000 | 6.115 | 218.525 | 0.000 | 5.215 |
| | Roica | 0.600 | 0.918 | 0.224 | 0.544 | 0.906 | 0.187 | 0.512 | 0.904 | 0.160 |
| | Roica-cv | 0.626 | 0.898 | 0.241 | 0.566 | 0.876 | 0.200 | 0.532 | 0.888 | 0.172 |
| PDC | Original | 125.669 | 0.000 | 2.825 | 125.104 | 0.000 | 2.332 | 124.524 | 0.000 | 1.983 |
| | Roica | 0.587 | 0.922 | 0.223 | 0.533 | 0.908 | 0.186 | 0.503 | 0.914 | 0.160 |
| | Roica-cv | 0.614 | 0.894 | 0.241 | 0.562 | 0.888 | 0.201 | 0.526 | 0.888 | 0.171 |
| RePPI | Original | 126.853 | 0.000 | 2.870 | 126.575 | 0.000 | 2.378 | 126.304 | 0.000 | 2.030 |
| | Roica | 0.588 | 0.920 | 0.223 | 0.532 | 0.906 | 0.186 | 0.503 | 0.906 | 0.160 |
| | Roica-cv | 0.624 | 0.872 | 0.239 | 0.553 | 0.878 | 0.198 | 0.527 | 0.876 | 0.171 |
| Supervised | | 0.658 | 0.910 | 0.287 | 0.588 | 0.910 | 0.240 | 0.565 | 0.908 | 0.205 |

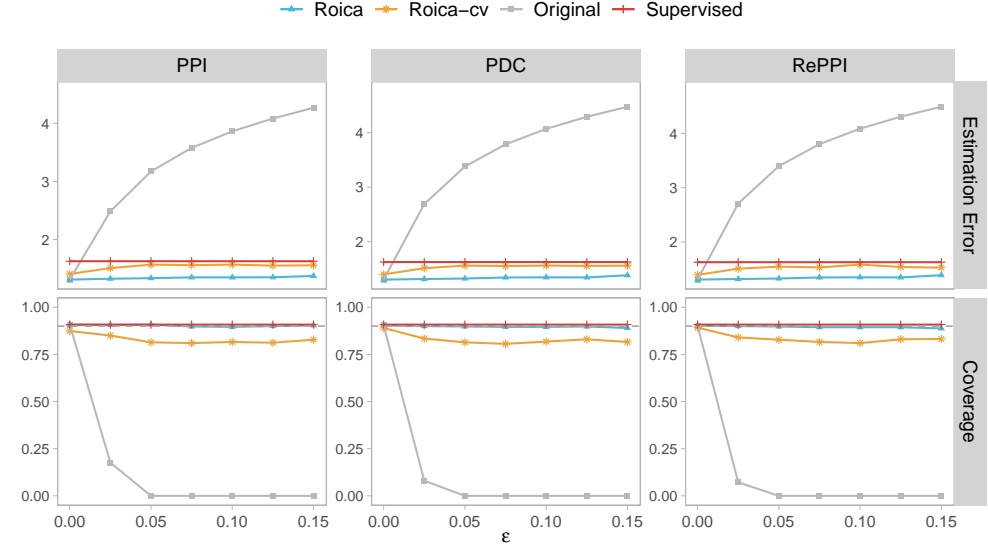

Figure 11: Empirical results of estimation error, coverage and the width of $90\%$ confidence intervals under aggressive attack with varying $\epsilon$ when $n = 500$ for the gene expression level estimation.

## D.2 Cross-Validation for Estimation of the Contamination level

Since the true contamination level $\epsilon_*$ is typically unknown in practice, we employ a cross-validation (CV) procedure on the labeled data to select an optimal value $\hat{\epsilon}$ as mentioned in Section 3.3. Here, we present the clear algorithmic steps in Algorithm 2.

---

**Algorithm 1** Filtering Algorithm

---

**Require:** Corrupted dataset $D_N = \{\tilde{X}_1, \tilde{X}_2, \cdots, \tilde{X}_N\}$, threshold for termination $\xi$
**Ensure:** Filtered mean estimate $\mathbb{E}_{w^{(k)}}[\tilde{X}]$
1: Initialize $w_i^{(0)} = 1/N$ for all $i \in [N]$
2: **for** $k = 0, 1, \ldots$ **do**
3:     **if** $\|\mathrm{Cov}_{w^{(k)}}(\tilde{X})\|_2 \le \xi$ **then**
4:         **return** $\mathbb{E}_{w^{(k)}}[\tilde{X}] = \sum_{i=1}^m w_i^{(k)} \tilde{X}_i$
5:     **else**
6:         **for** $i = 1$ to $N$ **do**
7:             Compute $\phi_i^{(k)} = \phi(w^{(k)}; \tilde{X}_i)$
8:             Update $w_i^{(k+1)} = w_i^{(k)} \cdot \left( 1 - \frac{\phi_i^{(k)}}{\max_{j \in [N]} \phi_j^{(k)}} \right)$
9:             Normalize: $w^{(k+1)} = \mathrm{Proj}_{\Delta_N}^{KL}(w^{(k+1)}) = w^{(k+1)} / \sum_{i=1}^N w_i^{(k+1)}$
10:           Remove samples with $w_i^{(k+1)} = 0$

---

**Algorithm 2** Cross-Validation for Contamination Level $\epsilon$ in $M$-Estimation

---

**Require:** Labeled data $\mathcal{D} = \{(X_i, Y_i)\}_{i=1}^n$, candidate set $\mathcal{E} = \{\epsilon_1, \epsilon_2, \cdots, \epsilon_M\}$, number of folds $K$, loss function $\ell_\theta(X, Y)$
**Ensure:** Optimal contamination level $\hat{\epsilon}$
1: Randomly partition $\mathcal{D}$ into $K$ folds: $\{\mathcal{D}_k\}_{k=1}^K$
2: Initialize $\mathrm{CVLoss}[\epsilon] \leftarrow 0$ for all $\epsilon \in \mathcal{E}$
3: **for** $\epsilon \in \mathcal{E}$ **do**
4:     **for** $k = 1$ to $K$ **do**
5:         Set training data: $\mathcal{D}_{\text{train}} \leftarrow \mathcal{D} \setminus \mathcal{D}_k$
6:         Set validation data: $\mathcal{D}_{\text{valid}} \leftarrow \mathcal{D}_k$
7:         Run robust estimation: $\theta_{-k}^{\mathcal{R}(\epsilon)} \leftarrow Roica$ procedure on $\mathcal{D}_{\text{train}}$ ▷ Using Algorithm 1 with $\epsilon$
8:         **for** $(X, Y) \in \mathcal{D}_{\text{valid}}$ **do**
9:             $\mathrm{CVLoss}[\epsilon] \leftarrow \mathrm{CVLoss}[\epsilon] + \ell_{\theta_{-k}^{\mathcal{R}(\epsilon)}}(X, Y)$
10: $\hat{\epsilon} \leftarrow \arg\min_{\epsilon \in \mathcal{E}} \mathrm{CVLoss}[\epsilon]$
11: **return** $\hat{\epsilon}$

---

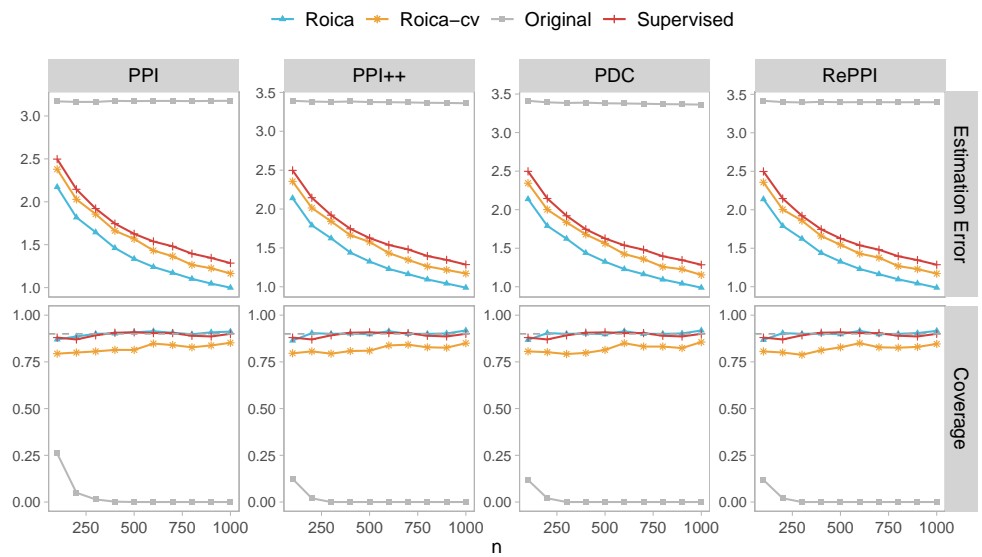

Figure 12: Empirical results of estimation error, coverage and the width of $90\%$ confidence intervals under moderate attack with varying $n$ when $\epsilon = 0.05$ for the gene expression level estimation.

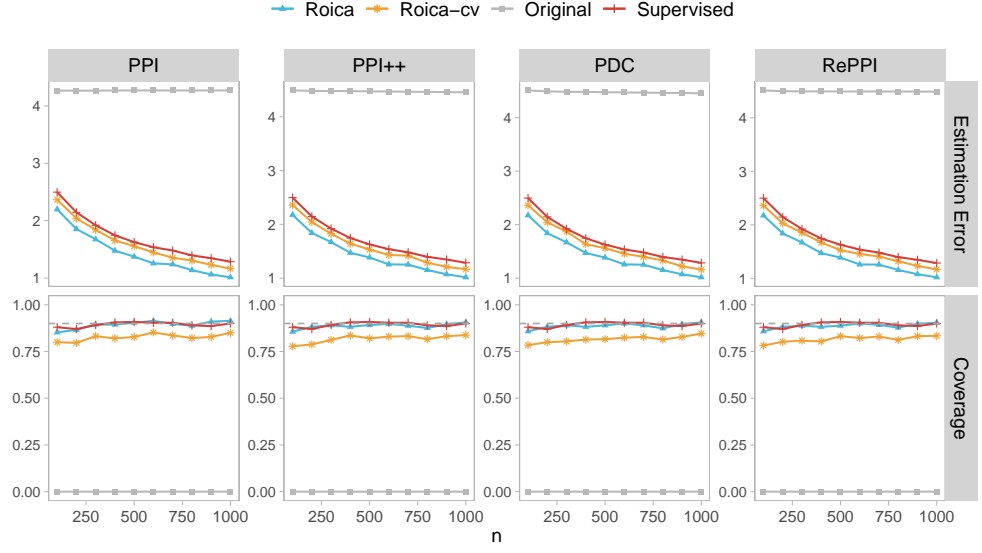

Figure 13: Empirical results of estimation error, coverage and the width of $90\%$ confidence intervals under aggressive attack with varying $n$ when $\epsilon = 0.15$ for the gene expression level estimation.

Table 12: Empirical results of estimation error, coverage and the width of $90\%$ confidence intervals under aggressive attack for $n = 500$ for the gene expression level estimation.

| Method | | $\epsilon = 0$ Error | $\epsilon = 0$ Coverage | $\epsilon = 0$ Width | $\epsilon = 0.05$ Error | $\epsilon = 0.05$ Coverage | $\epsilon = 0.05$ Volume | $\epsilon = 0.15$ Error | $\epsilon = 0.15$ Coverage | $\epsilon = 0.15$ Width |
|---|---|---|---|---|---|---|---|---|---|---|
| PPI | Original | 0.037 | 0.904 | 0.156 | 0.467 | 0.000 | 0.157 | 1.406 | 0.000 | 0.157 |
| | Roica | 0.037 | 0.904 | 0.156 | 0.037 | 0.892 | 0.157 | 0.037 | 0.896 | 0.157 |
| | Roica-cv | 0.041 | 0.876 | 0.156 | 0.039 | 0.888 | 0.157 | 0.040 | 0.864 | 0.157 |
| PPI++ | Original | 0.037 | 0.896 | 0.154 | 0.573 | 0.000 | 0.154 | 1.723 | 0.000 | 0.154 |
| | Roica | 0.037 | 0.896 | 0.154 | 0.037 | 0.896 | 0.154 | 0.038 | 0.888 | 0.154 |
| | Roica-cv | 0.041 | 0.878 | 0.154 | 0.039 | 0.882 | 0.154 | 0.041 | 0.876 | 0.154 |
| PDC | Original | 0.037 | 0.896 | 0.154 | 0.575 | 0.000 | 0.154 | 1.730 | 0.000 | 0.154 |
| | Roica | 0.037 | 0.896 | 0.154 | 0.037 | 0.896 | 0.154 | 0.038 | 0.888 | 0.154 |
| | Roica-cv | 0.041 | 0.882 | 0.154 | 0.038 | 0.882 | 0.154 | 0.041 | 0.856 | 0.154 |
| RePPI | Original | 0.037 | 0.892 | 0.154 | 0.585 | 0.000 | 0.154 | 1.759 | 0.000 | 0.154 |
| | Roica | 0.037 | 0.892 | 0.154 | 0.037 | 0.892 | 0.154 | 0.038 | 0.884 | 0.154 |
| | Roica-cv | 0.041 | 0.882 | 0.154 | 0.038 | 0.882 | 0.154 | 0.040 | 0.868 | 0.154 |
| Supervised | | 0.051 | 0.906 | 0.212 | 0.050 | 0.908 | 0.213 | 0.051 | 0.904 | 0.213 |

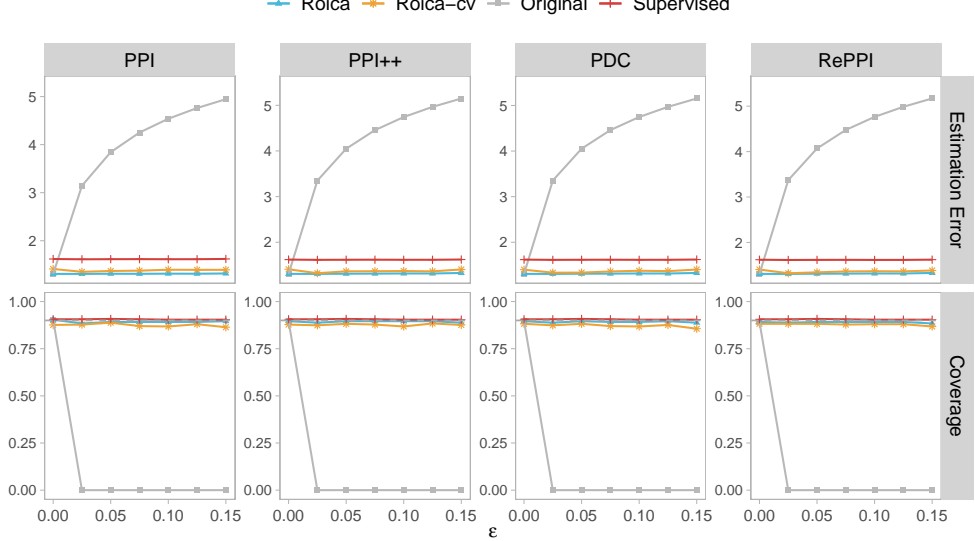

Figure 14: Empirical results of estimation error, coverage and the width of $90\%$ confidence intervals under severe attack with varying $\epsilon$ when $n = 500$ for the gene expression level estimation.

Table 13: Empirical results of estimation error, coverage and the volume of 90% confidence regions under higher contamination ratios when $b = 20$ for linear regression.

| Method | | $\epsilon = 0.2$ | | | $\epsilon = 0.3$ | | | $\epsilon = 0.4$ | | |
|---|---|---|---|---|---|---|---|---|---|---|
| | | Error | Coverage | Volume | Error | Coverage | Volume | Error | Coverage | Volume |
| PPI | Original | 10.353 | 0.000 | 4.944 | 10.360 | 0.000 | 4.979 | 10.363 | 0.000 | 5.003 |
| | Roica | 0.058 | 0.906 | 0.003 | 0.059 | 0.904 | 0.003 | 0.060 | 0.904 | 0.003 |
| | Roica-cv | 0.063 | 0.886 | 0.004 | 0.065 | 0.880 | 0.004 | 0.066 | 0.874 | 0.004 |
| PPI++ | Original | > 100 | 0.000 | > 100 | > 100 | 0.000 | > 100 | > 100 | 0.000 | > 100 |
| | Roica | 0.058 | 0.910 | 0.003 | 0.059 | 0.910 | 0.003 | 0.060 | 0.904 | 0.003 |
| | Roica-cv | 0.065 | 0.884 | 0.004 | 0.063 | 0.884 | 0.004 | 0.065 | 0.888 | 0.004 |
| PDC | Original | > 100 | 0.000 | > 100 | > 100 | 0.000 | > 100 | > 100 | 0.000 | > 100 |
| | Roica | 0.067 | 0.874 | 0.003 | 0.062 | 0.896 | 0.003 | 0.075 | 0.876 | 0.004 |
| | Roica-cv | 0.067 | 0.870 | 0.004 | 0.066 | 0.884 | 0.004 | 0.076 | 0.866 | 0.004 |
| RePPI | Original | > 100 | 0.000 | > 100 | > 100 | 0.000 | > 100 | > 100 | 0.000 | > 100 |
| | Roica | 0.059 | 0.890 | 0.003 | 0.063 | 0.874 | 0.003 | 0.089 | 0.852 | 0.004 |
| | Roica-cv | 0.063 | 0.878 | 0.003 | 0.065 | 0.864 | 0.004 | 0.066 | 0.856 | 0.004 |
| Supervised | | 0.076 | 0.892 | 0.007 | 0.076 | 0.892 | 0.007 | 0.076 | 0.892 | 0.007 |

Table 14: Empirical results of estimation error, coverage and the volume of 90% confidence regions under adversarial attack when $\delta = 20$ for linear regression.

| Method | | $\epsilon = 0$ | | | $\epsilon = 0.05$ | | | $\epsilon = 0.15$ | | |
|---|---|---|---|---|---|---|---|---|---|---|
| | | Error | Coverage | Volume | Error | Coverage | Volume | Error | Coverage | Volume |
| PPI | Original | 0.058 | 0.914 | 0.003 | 5.629 | 0.000 | 0.831 | 5.867 | 0.000 | 0.938 |
| | Roica | 0.058 | 0.914 | 0.003 | 0.058 | 0.900 | 0.003 | 0.058 | 0.904 | 0.003 |
| | Roica-cv | 0.064 | 0.884 | 0.004 | 0.064 | 0.874 | 0.004 | 0.063 | 0.882 | 0.004 |
| PPI++ | Original | 0.058 | 0.918 | 0.003 | 84.331 | 0.000 | > 100 | > 100 | 0.000 | > 100 |
| | Roica | 0.058 | 0.918 | 0.003 | 0.058 | 0.912 | 0.003 | 0.058 | 0.910 | 0.003 |
| | Roica-cv | 0.062 | 0.894 | 0.004 | 0.063 | 0.898 | 0.004 | 0.063 | 0.898 | 0.004 |
| PDC | Original | 0.058 | 0.912 | 0.003 | 93.082 | 0.000 | > 100 | > 100 | 0.000 | > 100 |
| | Roica | 0.058 | 0.912 | 0.003 | 0.065 | 0.872 | 0.003 | 0.062 | 0.888 | 0.003 |
| | Roica-cv | 0.062 | 0.896 | 0.004 | 0.066 | 0.870 | 0.004 | 0.066 | 0.878 | 0.004 |
| RePPI | Original | 0.055 | 0.908 | 0.003 | 93.287 | 0.000 | > 100 | > 100 | 0.000 | > 100 |
| | Roica | 0.055 | 0.908 | 0.003 | 0.057 | 0.894 | 0.003 | 0.057 | 0.890 | 0.003 |
| | Roica-cv | 0.060 | 0.878 | 0.003 | 0.063 | 0.878 | 0.003 | 0.063 | 0.876 | 0.003 |
| Supervised | | 0.076 | 0.892 | 0.007 | 0.076 | 0.892 | 0.007 | 0.076 | 0.892 | 0.007 |

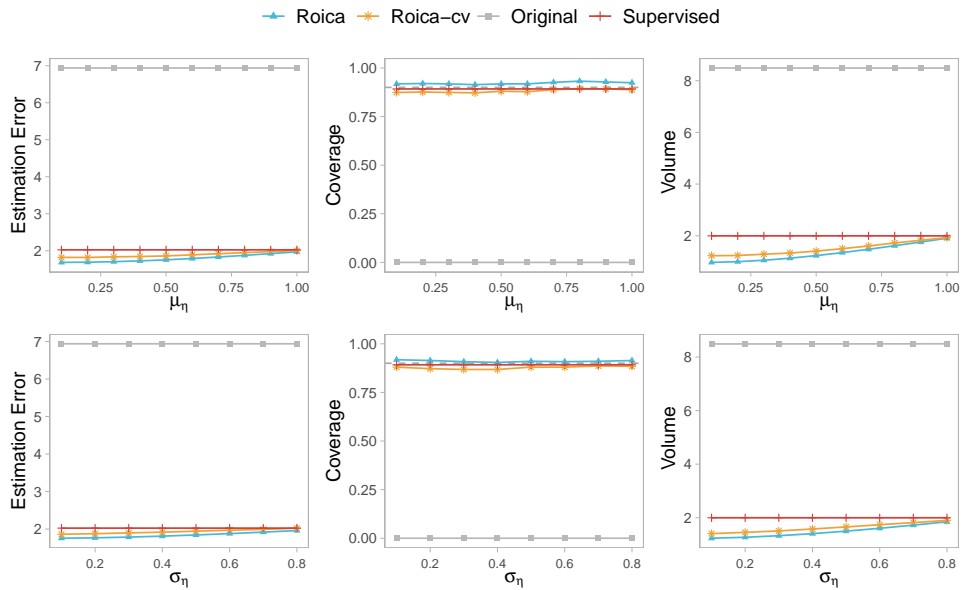

Figure 15: Empirical results of estimation error, coverage and the volume of $90\%$ confidence regions with varying $\mu_\eta$ and $\sigma_\eta$ when $\epsilon = 0.1$.

