# OpenReview forum: "Robust Prediction-Powered Inference under Data Corruption"
_ICLR.cc/2026/Conference — ICLR 2026 Conference Desk Rejected Submission_

### Official Review · Reviewer_LWQZ · 2025-10-30

**Soundness:** 3
**Presentation:** 2
**Contribution:** 3
**Rating:** 4
**Confidence:** 3

**Summary:**

This paper proposes Roica (Robust prediction-powered inference with calibration), a method that enhances the Prediction-Powered Inference (PPI) framework. It substitutes the step of averaging the "imputed gradients" for unlabeled samples with robust mean estimation. Furthermore, it employs cross-validation on the labeled data to select the contamination level, aiming to enable reliable semi-supervised estimation and inference even when the unlabeled set is subject to contamination.

**Strengths:**

This article mainly solves the problem of contaminated data. The author proposes a robust method that can also use unlabeled data when contaminated data is limited.

**Weaknesses:**

I have several points:
1. The paper claims that the proposed method improves robustness. My question is, does this refer only to an improvement in consistency, or does it also include gains in estimation efficiency? If it's only about consistency, why can't we just use the labeled data directly?
2. Would you to clarify the meaning of the $\epsilon=\Theta(\epsilon_*+\log(1/\tau)/N)$ symbol in line 257. My understanding was that $\Theta$ denotes the parameter space, so its usage here with an equals sign seems unusual. Could you please explain its specific meaning in this context?
3.To enhance clarity, I suggest providing further explanation of Assumptions 1 and 2. This would facilitate a deeper understanding of their role and justification for the reader.
4.The significance of the result in Theorem 1 could be further highlighted by providing a clearer discussion. It would be very helpful to contextualize it against existing methods—clarifying how it aligns with or diverges from related work would make the specific advantages and novel contributions of your approach more apparent to the reader.
5. The performance of the proposed method appears to rely on the accuracy of the pre-trained model $f(\cdot)$. Could you comment on the robustness of your approach to potential inaccuracies in this model? Specifically, an analysis of its sensitivity to $f(\cdot)$ would be highly informative.
6. Both the lemma and the theorem demonstrate that the estimation is related to the dimension of the parameters. However, the simulation only considers p=3, while n and N are particularly large. Could we represent more cases with relatively small n, such as 100, and relatively large p, such as 10 or 20?

**Questions:**

please see the weakness

---

> ### Author Response · Authors · 2025-11-21
> **Response to Reviewer LWQZ**
>
> We thank the reviewer for the constructive feedback. Below, we respond to each comment point-by-point, with the original text in a blockquote.
>
> > The paper claims that the proposed method improves robustness. My question is, does this refer only to an improvement in consistency, or does it also include gains in estimation efficiency? If it's only about consistency, why can't we just use the labeled data directly?
> The significance of the result in Theorem 1 could be further highlighted by providing a clearer discussion. It would be very helpful to contextualize it against existing methods—clarifying how it aligns with or diverges from related work would make the specific advantages and novel contributions of your approach more apparent to the reader.
> >
>
> We thank the reviewer for this critical question. The improvement refers to both consistency and estimation efficiency. As established in Theorem 1, which characterizes the asymptotic normal distribution of our estimator, the asymptotic variance of our robust estimator is strictly smaller than that of the supervised estimator.  For simplicity, here we assume $p = 1$ and consider the inference problem of the mean value of $Y$. When the pre-trained model provides a good approximation of the label $ Y $, i.e $ \Delta = f(X) - Y $ is small, we have $ \sigma_{\Delta}^2 \ll \sigma_{\nabla\ell_{\theta}(X,Y)}^2 $, where $ \sigma_{\Delta}^2 $ is the asymptotic variance of the Roica method by Theorem 1 and $ \sigma_{\nabla\ell_{\theta}(X,Y)}^2 $ is that of the supervised method. This theoretically guarantees that our method not only remains consistent under contamination but also delivers more efficient estimation, yielding smaller confidence regions. This is precisely why using only labeled data (the supervised estimator) is suboptimal --- it fails to exploit the unlabeled data to achieve this statistical efficiency gain.
>
> We also added the discussion for more general tasks after Theorem 1 in the revised version for better understanding.
>
>
>
>
> > To enhance clarity, I suggest providing further explanation of Assumptions 1 and 2. This would facilitate a deeper understanding of their role and justification for the reader.
> >
> Many thanks for your suggestions on the presentation of the theoretical part.     We add the following discussion on the role of these assumptions in the revised manuscript:
> - "Assumption 1 controls the spectral norm of the covariance matrix of the imputed gradients. This condition is fundamental to robust mean estimation (specifically the Filtering algorithm used in Roica), as it ensures that the variance of the clean data is bounded, allowing the algorithm to effectively identify outliers based on covariance inflation. Assumption 2 gathers standard regularity conditions for M-estimation \citep{van1998cambridge}. Specifically, (a), (c), and (d) ensure the parameter space is well-behaved and the moments exist for the central limit theorem; (b) requires the Hessian estimated from labeled data to be consistent, ensuring the validity of the one-step update; and (e) imposes Lipschitz continuity to guarantee the Donsker property, which is necessary to bound the stochastic error terms in the asymptotic expansion."
>
>
>
>
> > Would you to clarify the meaning of the symbol $\epsilon=\Theta(\epsilon_*+\log(1/\tau)/N)$ in line 257. My understanding was that $ \Theta $ denotes the parameter space, so its usage here with an equals sign seems unusual. Could you please explain its specific meaning in this context?
> >
>
> We thank the reviewer for catching this potential ambiguity. In this line, $ \Theta $ is used to denote asymptotic tight bounds (from algorithm analysis), meaning $ \epsilon $ is upper and lower bounded by $ \epsilon_*+\log(1/\tau)/N $ up to constant factors. It does not refer to the parameter space here. We clarify this definition in the revision of the main text.

---

> ### Author Response · Authors · 2025-11-21
> **Response to Reviewer LWQZ (cont'd)**
>
> > The performance of the proposed method appears to rely on the accuracy of the pre-trained model $ f(\cdot) $. Could you comment on the robustness of your approach to potential inaccuracies in this model? Specifically, an analysis of its sensitivity to $ f(\cdot) $ would be highly informative.
> >
>
> We thank the reviewer for this critical question regarding the robustness to model misspecification. We address this point from two perspectives:
>
> + Inherent Design of the PPI Framework: A key foundation of the standard Prediction-Powered Inference (PPI) framework, which our method builds upon, is that it is explicitly designed based on a fixed, misspecified model. The core PPI algorithm does not require $ f(\cdot) $ to be perfectly accurate; instead, it uses the labeled data to **correct for the bias** of this potentially inaccurate model. Therefore, our method inherits this robustness to a constant prediction bias directly from the PPI paradigm.
> + Empirical Sensitivity Analysis: To quantitatively assess the sensitivity to variations in the prediction model, we have conducted a dedicated analysis in the linear regression setting. We systematically varied the parameters of the predictive model $ f(X) = X^\top \beta_1 + (X^2)^\top \beta_2 + \eta, \quad \eta \sim \mathcal{N}(-\mu_\eta, \sigma_\eta^2) $. The results presented in Appendix C.6 of the revised manuscript, illustrate how the performance of our method changes with the mean $ \mu_\eta $ (capturing systematic bias) and the variance $ \sigma_\eta $ (capturing random inaccuracy) of the predictor.
>
> Indeed, as theoretically anticipated, the performance of all methods is influenced by increasing model bias($\mu_\eta$) and variance ($\sigma_\eta$). From Figure 15 of the revised manuscript, a consistent observation is that Roica (and Roica-cv) robustly maintains valid coverage across all perturbation levels.
>
> Concurrently, as the predictor's inaccuracy grows whether through systematic bias or random noise, the estimation error and the volume of the confidence region for our method gradually decrease. This phenomenon is intuitive: as the model predictions become less reliable, the information they provide becomes inherently less certain.
>
>
> > Both the lemma and the theorem demonstrate that the estimation is related to the dimension of the parameters. However, the simulation only considers p=3, while n and N are particularly large. Could we represent more cases with relatively small n, such as 100, and relatively large p, such as 10 or 20?
> >
>
> We thank the reviewer for this excellent suggestion. We agree that investigating the method's performance in higher-dimensional settings with smaller labeled sample sizes is crucial for demonstrating its broad applicability. In direct response to this comment, we have conducted new experiments with a smaller labeled set size ($ n = 100$) and larger dimension ($ p = 10 $). The table below exhibits the empirical results of the original PPI and our Roica methods with varying contamination levels. While the high dimension and small labeled set size do reduce estimation precision and inferential efficiency (as expected), our results clearly show that Roica and Roica-cv consistently outperform both the supervised estimator and the original baselines across all conditions tested.
>
> | Method | $\epsilon = 0$ | | | | $\epsilon = 0.05$ | | | | $\epsilon = 0.1$ | | |
> |:---|:---|:---|:---|:---|:---|:---|:---|:---|:---|:---|:---|
> | | Error | Coverage | Volume | | Error | Coverage | Volume | | Error | Coverage | Volume |
> | Original | 0.364 | 0.732 | 0.001 | | 18.923 | 0.000 | >100 | | 18.976 | 0.000 | >100 |
> | Roica | 0.364 | 0.732 | 0.001 | | 0.411 | 0.676 | 0.001 | | 0.411 | 0.684 | 0.001 |
> | Roica-cv | 0.399 | 0.658 | 0.003 | | 0.486 | 0.626 | 0.011 | | 0.487 | 0.620 | 0.011 |
> | Supervised | 0.635 | 0.554 | 0.054 | | 0.635 | 0.554 | 0.054 | | 0.635 | 0.554 | 0.054 |

---

> > ### Comment · Reviewer_LWQZ · 2025-11-27
> >
> > I thank the authors for addressing my questions. I am currently maintaining my score, and there is a possibility of increasing it. The final decision will require discussion with other reviewers.

---

> > > ### Author Response · Authors · 2025-11-28
> > > **Response to Reviewer LWQZ**
> > >
> > > We appreciate the time and effort invested by the reviewer in carefully evaluating our work. Thank you for recognizing our core contribution—proposing a robust inference procedure that can securely leverage large-scale unlabeled data even when these data are corrupted. We also thank you for the helpful suggestions on the writing of the manuscript.

---

### Official Review · Reviewer_XDuQ · 2025-10-31

**Soundness:** 2
**Presentation:** 3
**Contribution:** 2
**Rating:** 2
**Confidence:** 4

**Summary:**

This paper addresses robustness issues in Prediction-Powered Inference (PPI), which uses model-imputed labels for semi-supervised inference. Standard PPI assumes the unlabeled covariate distribution matches the labeled one and breaks under distribution shift or adversarial contamination. The proposed Roica framework adds a robust mean estimator to the imputation-and-calibration step and uses cross-validation on labeled data to choose a contamination level. The authors provide asymptotic theory and demonstrate advantages over prior PPI variants in synthetic and real-data settings (ACS income data and yeast promoter gene-expression).

**Strengths:**

1. Theory appears sound: consistency and asymptotic normality under contamination.
2. The robust mean-estimation method is principled and well-studied in high-dimensional statistics.

**Weaknesses:**

1. The motivation for addressing corruption in PPI appears largely hypothetical. The paper does not demonstrate that the data-contamination scenarios it considers represent genuine bottlenecks in real semi-supervised inference workflows, raising concerns that the problem setting is constructed primarily to justify the proposed method. All the real data applications are made up by simulations to fit the problem that the author described. I would suggest to have more convincing motivations to motivate the paper.
2. The core contribution is swapping in a known robust mean estimator. Most structural elements (bias-correction form, one-step Newton refinement) are directly inherited from prior PPI work.
3. The contamination mechanism (simple additive shifts) may not reflect realistic mis-specifications like conditional shift, label-related corruption, or adversarial errors targeted at the model’s blind spots. How does Roica perform with those other situations?

**Questions:**

1. See weakness
2. Can the contamination estimation step over-shrink toward supervised estimates and reduce PPI gains?
3. Would Roica still help when model predictions are biased in-distribution, not only on corrupted points?

---

> ### Author Response · Authors · 2025-11-21
> **Response to Reviewer XDuQ**
>
> We are grateful for the reviewer's valuable insights and have prepared a detailed response to all points raised by the reviewer. (The original comments are included in italics.)
>
> > _The motivation for addressing corruption in PPI appears largely hypothetical......I would suggest to have more convincing motivations to motivate the paper._
>
> We thank the reviewer for challenging the motivation of the work. We agree that strictly 'adversarial' corruption is a specific worst-case scenario. Our primary motivation also stems from the fundamental challenge of **Out-of-Distribution (OOD) generalization**. In real-world workflows, labeled data is often carefully curated and cleaned (representing the target distribution), whereas unlabeled data is frequently collected in the 'wild' (e.g., web-scraped data, user logs, or data from different time periods). This discrepancy inevitably introduces covariate shifts and 'natural' contamination in the unlabeled set. Standard PPI methods assume perfect distributional alignment and fail under these shifts. We model this practical OOD problem using the contamination framework to provide rigorous guarantees.
>
> Based on your advice, we also added more discussion on the motivation of the adversarial-robust modeling and the relation with the OOD generalization task in the introduction.
>
> > _The core contribution is swapping in a known robust mean estimator. Most structural elements (bias-correction form, one-step Newton refinement) are directly inherited from prior PPI work._
>
> We thank the reviewer for this observation. We openly acknowledge that our method leverages the elegant algorithmic backbone of Prediction-Powered Inference (PPI). However, we respectfully argue that the integration of robust estimation is far from a trivial 'swap.' Standard PPI relies on the strict assumption of covariate homogeneity; when this is violated (e.g., in OOD or contaminated settings), the original framework collapses, yielding invalid inference. Our contribution is to rigorously discuss and solve this issue. Furthermore, we introduce a novel cross-validation mechanism (Roica-cv) to empirically select the contamination level, a practical necessity that is absent in standard robust estimation literature.
>
> > _The contamination mechanism (simple additive shifts) may not reflect realistic mis-specifications like conditional shift, label-related corruption, or adversarial errors targeted at the model’s blind spots. How does Roica perform with those other situations?_
>
> We thank the reviewer for this important question. A similar recommendation was also made by Reviewer m9vm. Regarding conditional shift and label-related corruption, these are difficult to simulate in our semi-supervised setting since the true labels $Y$ for unlabeled data are unobserved.
>
> In response to the need for more realistic adversarial testing, we employ a sophisticated contamination scheme inspired by [1, 2]. This approach is designed to perturb covariates along their first principal component ($ \tilde{X}^o_i = \tilde{X}_i + \delta v_1 $), thereby directly attacking the data's fundamental structure, where $ v_1 $ is the leading principal component direction of the unlabeled dataset and $\delta$ controls the intensity. The ensuing results for linear regression (added in Appendix C.5 of the revised manuscript), displayed in the table below, provide conclusive evidence that our method (Roica) sustains robust performance in this demanding adversarial context and surpasses all comparative methods.
>
> |||$\epsilon=0$||||$\epsilon=0.05$||||$\epsilon=0.15$|||
> |:---|:---|:---|:---|:---|:---|:---|:---|:---|:---|:---|:---|:---|
> |Method||Error|Coverage|Volume||Error|Coverage|Volume||Error|Coverage|Volume|
> |PPI|Original|0.058|0.914|0.003||5.629|0.000|0.831||5.867|0.000|0.938|
> ||Roica|0.058|0.914|0.003||0.058|0.900|0.003||0.058|0.904|0.003|
> ||Roica-cv|0.064|0.884|0.004||0.064|0.874|0.004||0.063|0.882|0.004|
> |PPI++|Original|0.058|0.918|0.003||84.331|0.000|>100||>100|0.000|>100|
> ||Roica|0.058|0.918|0.003||0.058|0.912|0.003||0.058|0.910|0.003|
> ||Roica-cv|0.062|0.894|0.004||0.063|0.898|0.004||0.063|0.898|0.004|
> |PDC|Original|0.058|0.912|0.003||93.082|0.000|>100||>100|0.000|>100|
> ||Roica|0.058|0.912|0.003||0.065|0.872|0.003||0.062|0.888|0.003|
> ||Roica-cv|0.062|0.896|0.004||0.066|0.870|0.004||0.066|0.878|0.004|
> |RePPI|Original|0.055|0.908|0.003||93.287|0.000|>100||>100|0.000|>100|
> ||Roica|0.055|0.908|0.003||0.057|0.894|0.003||0.057|0.890|0.003|
> ||Roica-cv|0.060|0.878|0.003||0.063|0.878|0.003||0.063|0.876|0.003|
> |Supervised||0.076|0.892|0.007||0.076|0.892|0.007||0.076|0.892|0.007|
>
>
> [1] G. Li et al., "Adversarial Attacks on Principal Component Analysis," in _International Conference on Machine Learning_, 2020.
>
> [2] I. Diakonikolas and D. M. Kane, "Robust Estimation in High Dimensions: A Survey," _Foundations and Trends® in Machine Learning_, vol. 14, no. 1-2, pp. 1–224, 2021.

---

> > ### Author Response · Authors · 2025-11-21
> > **Response to Reviewer XDuQ (cont'd)**
> >
> > > _Can the contamination estimation step over-shrink toward supervised estimates and reduce PPI gains?_
> >
> > We thank the reviewer for raising this important point. We address the issue of 'negative learning' (where unlabeled data hurts performance) by explicitly including the supervised estimator (represented by the $\epsilon=1$ case) in our cross-validation candidate set. Consequently, if the unlabeled data is heavily contaminated or shifted such that it would degrade estimation quality, the cross-validation procedure will identify this and select the supervised estimate. It guarantees that our method remains robust and does not underperform the supervised baseline.
> >
> > As shown in Figure 3 and Table 2 in the manuscript, Roica-cv (estimating $ \epsilon $ via cross-validation) consistently achieves confidence regions that are significantly smaller than the supervised estimator while maintaining valid coverage.
> >
> > > _Would Roica still help when model predictions are biased in-distribution, not only on corrupted points?_
> >
> > We thank the reviewer for this insightful question. Indeed, the standard PPI framework (including PPI and the variants) is precisely designed under the assumption that model predictions are biased, and it systematically corrects for this bias using labeled data. This bias correction is a fundamental strength of the PPI paradigm. Roica provides additional robustness against data contamination while preserving PPI's predition-bias correction capabilities.

---

> > > ### Comment · Reviewer_XDuQ · 2025-11-27
> > >
> > > I thank the authors for the detailed response. I would also encourage the authors for finding motivating examples to illustrate that the setting that the authors' method trying to tackle is relevant and important.

---

> > > > ### Author Response · Authors · 2025-11-28
> > > > **Response to Reviewer XDuQ**
> > > >
> > > > We sincerely thank the reviewer for this insightful comment and constructive suggestion. We fully agree that demonstrating the practical relevance and importance of our methodological setting through compelling real-world examples would significantly strengthen the impact of our work. The reviewer's guidance is greatly appreciated and will importantly shape the direction of our future research in this area.

---

### Official Review · Reviewer_m9vm · 2025-11-01

**Soundness:** 2
**Presentation:** 2
**Contribution:** 2
**Rating:** 2
**Confidence:** 5

**Summary:**

This paper introduces a framework designed to enhance the robustness of Prediction-Powered Inference (PPI) in semi-supervised learning settings when the unlabeled data are contaminated or drawn from a shifted distribution. The authors propose incorporating a robust mean estimator into the imputation–calibration step of PPI to mitigate the adverse effects of data corruption. In addition, a cross-validation procedure is developed to select the contamination proportion parameter. The paper establishes asymptotic normality for the resulting estimator and presents empirical evaluations on synthetic data and two real-world datasets, demonstrating improved robustness compared with several existing PPI variants.

**Strengths:**

Conceptual simplicity and generality: The proposed approach integrates robust estimation principles into PPI in a way that is conceptually straightforward and potentially applicable to various models.

Theoretical consistency: The asymptotic properties (consistency and asymptotic normality) of the proposed estimator are formally established under standard regularity conditions.

**Weaknesses:**

Limited novelty: The methodological contribution is largely an application of known robust mean estimation techniques within the PPI framework. The main idea—replacing the sample mean in PPI by a robust alternative—is conceptually incremental and lacks substantive methodological innovation.

Marginal theoretical contribution: The theoretical results mainly rely on standard asymptotic arguments for M-estimation and established results from the robust statistics literature. The paper does not develop new theoretical insights specific to the interaction between robustness and semi-supervised inference.

Incomplete methodological clarity: The implementation details of the robust weighting scheme (Eq. 2) are insufficiently described. The optimization problem may be computationally demanding for large unlabeled datasets, yet no complexity analysis or practical guidance is provided.

Simplistic experimental design: The contamination mechanisms used in simulations (additive shifts, replacement of a small fraction of samples) are stylized and do not convincingly capture the complexities of real-world distributional shifts. Moreover, the empirical analysis focuses narrowly on PPI-type baselines rather than more general robust SSL or domain adaptation methods.

Overstated empirical claims: While Roica achieves favorable coverage and smaller confidence regions under artificial contamination, the results do not clearly demonstrate improved real-world predictive or inferential performance beyond these metrics.

Expositional issues: The presentation is notation-heavy and sometimes opaque. Key algorithmic steps (e.g., the filtering procedure and cross-validation routine) would benefit from clearer pseudocode or schematic illustration.

**Questions:**

How does the proposed robust mean estimation procedure differ in practice from existing filtering-based robust estimators? Is any modification specifically tailored to the PPI setting?

How stable is the cross-validation procedure for estimating the contamination level, particularly when the labeled sample size is small?

What are the computational and scalability properties of solving Eq. (2)? How does performance degrade with increasing dimensionality or unlabeled sample size?

The theoretical results assume mild contamination. Can the method tolerate higher contamination levels, and how would it perform when this assumption fails?

How would Roica compare against robust domain adaptation or covariate-shift correction methods, which also address distributional heterogeneity?

---

> ### Author Response · Authors · 2025-11-21
> **Response to Reviewer m9vm**
>
> We thank the reviewer for their thorough review and constructive suggestions. We have carefully considered all the comments and provide our detailed responses below. (The original comments are included in italics.)
>
> > _Incomplete methodological clarity: The implementation details of the robust weighting scheme (Eq. 2) are insufficiently described. The optimization problem may be computationally demanding for large unlabeled datasets, yet no complexity analysis or practical guidance is provided._
> >
> > _What are the computational and scalability properties of solving Eq. (2)? How does performance degrade with increasing dimensionality or unlabeled sample size?_
>
> We are grateful to the reviewer for highlighting the need for greater methodological clarity and computational discussion around our weighting scheme (Eq. (2)). We have substantially expanded a detailed algorithm of Filtering in Appendix D.1. to clarify the implementation. Regarding computational demand, the proposed robust weighting scheme has a computational complexity of $\tilde{O}(\epsilon N^2 d^3)$ [1], where $\tilde{O}(\cdot)$ hides logarithmic factors. The computation time can be further improved to near linear with more careful algorithm design [2].
>
> > _Simplistic experimental design: The contamination mechanisms used in simulations (additive shifts, replacement of a small fraction of samples) are stylized and do not convincingly capture the complexities of real-world distributional shifts._
>
> We thank the reviewer for this insightful comment regarding the realism of our contamination mechanisms. A similar suggestion has been proposedraised by Reviewer XDuQ. Inspired by [3, 4], we introduce a more challenging **adversarial contamination** scheme aimed at corrupting the fundamental structure of the covariate space ($ X $) in the revised version.  The specific methodology is as follows: randomly selected samples are perturbed along the direction of the dataset's first principal component, formulated as $ \tilde{X}^o_i = \tilde{X}_i + \delta v_1 $
> where $ v_1 $ represents the first principal component direction of the sample covariance of the unlabeled covariates, and $\delta$ denotes the perturbation intensity.  The results for linear regression presented in the table below (added in Appendix C.5. of the revised manuscript) conclusively demonstrate that our method (Roica) maintains robust performance even under this stringent adversarial setting, outperforming the comparison methods.
>
> | | | $\epsilon = 0$ | | | | $\epsilon = 0.05$ | | | | $\epsilon = 0.15$ | | |
> |:---|:---|:---|:---|:---|:---|:---|:---|:---|:---|:---|:---|:---|
> | Method | | Error | Coverage | Volume | | Error | Coverage | Volume | | Error | Coverage | Volume |
> | PPI | Original | 0.058 | 0.914 | 0.003 | | 5.629 | 0.000 | 0.831 | | 5.867 | 0.000 | 0.938 |
> | | Roica | 0.058 | 0.914 | 0.003 | | 0.058 | 0.900 | 0.003 | | 0.058 | 0.904 | 0.003 |
> | | Roica-cv | 0.064 | 0.884 | 0.004 | | 0.064 | 0.874 | 0.004 | | 0.063 | 0.882 | 0.004 |
> | PPI++ | Original | 0.058 | 0.918 | 0.003 | | 84.331 | 0.000 | >100 | | >100 | 0.000 | >100 |
> | | Roica | 0.058 | 0.918 | 0.003 | | 0.058 | 0.912 | 0.003 | | 0.058 | 0.910 | 0.003 |
> | | Roica-cv | 0.062 | 0.894 | 0.004 | | 0.063 | 0.898 | 0.004 | | 0.063 | 0.898 | 0.004 |
> | PDC | Original | 0.058 | 0.912 | 0.003 | | 93.082 | 0.000 | >100 | | >100 | 0.000 | >100 |
> | | Roica | 0.058 | 0.912 | 0.003 | | 0.065 | 0.872 | 0.003 | | 0.062 | 0.888 | 0.003 |
> | | Roica-cv | 0.062 | 0.896 | 0.004 | | 0.066 | 0.870 | 0.004 | | 0.066 | 0.878 | 0.004 |
> | RePPI | Original | 0.055 | 0.908 | 0.003 | | 93.287 | 0.000 | >100 | | >100 | 0.000 | >100 |
> | | Roica | 0.055 | 0.908 | 0.003 | | 0.057 | 0.894 | 0.003 | | 0.057 | 0.890 | 0.003 |
> | | Roica-cv | 0.060 | 0.878 | 0.003 | | 0.063 | 0.878 | 0.003 | | 0.063 | 0.876 | 0.003 |
> | Supervised | | 0.076 | 0.892 | 0.007 | | 0.076 | 0.892 | 0.007 | | 0.076 | 0.892 | 0.007 |
>
> [1] Banghua Zhu, Lun Wang, Qi Pang, Shuai Wang, Jiantao Jiao, Dawn Song, Michael I. Jordan. Byzantine-Robust Federated Learning with Optimal Statistical Rates. Proceedings of The 26th International Conference on Artificial Intelligence and Statistics, PMLR 206:3151-3178, 2023.
>
> [2] Depersin, J. and Lecué, G., 2022. Robust sub-Gaussian estimation of a mean vector in nearly linear time. The Annals of Statistics, 50(1), pp.511-536.
>
> [3] G. Li et al., "Adversarial Attacks on Principal Component Analysis," in _International Conference on Machine Learning_, 2020.
>
> [4] I. Diakonikolas and D. M. Kane, "Robust Estimation in High Dimensions: A Survey," _Foundations and Trends® in Machine Learning_, vol. 14, no. 1-2, pp. 1–224, 2021.

---

> ### Author Response · Authors · 2025-11-21
> **Response to Reviewer m9vm (cont'd)**
>
> > _Overstated empirical claims: While Roica achieves favorable coverage and smaller confidence regions under artificial contamination, the results do not clearly demonstrate improved real-world predictive or inferential performance beyond these metrics._
>
> We appreciate the reviewer's point regarding real-world inferential performance. To clarify, the estimation error results (e.g., in Figure 3 / Table 2) directly address this concern. They demonstrate that our method not only achieves valid coverage with smaller confidence regions but also provides more accurate point estimates. This reduction in estimation error translates to improved inferential efficiency in practice, going beyond mere metric-driven coverage guarantees.
>
>
> > _Expositional issues: The presentation is notation-heavy and sometimes opaque. Key algorithmic steps (e.g., the Filtering procedure and cross-validation routine) would benefit from clearer pseudocode or schematic illustration._
>
> We thank the reviewer for this constructive suggestion to improve the exposition. In response, we have now included clear, step-by-step pseudocode for both the Filtering procedure and the cross-validation routine in Appendix D. We believe these additions significantly enhance the clarity and reproducibility of our work.
>
>
>
> > _How does the proposed robust mean estimation procedure differ in practice from existing Filtering-based robust estimators? Is any modification specifically tailored to the PPI setting?_
>
> One of the issue of the original robust mean estimator Filtering，is that it depends on the hyperparameter $ \epsilon $. The semi-supervised method we considered provides a feasible way to estimate it. With the proposed cross-validation procedure in this work, we are able to make the robust mean estimator fully data-driven.
>
> > _How stable is the cross-validation procedure for estimating the contamination level, particularly when the labeled sample size is small?_
>
> We thank the reviewer for this critical question. We acknowledge that estimation stability can be a concern with limited labeled data. To thoroughly evaluate this, we conducted extensive simulations with the small labeled sample size $ n = 200 $ and $ n=500 $. The results shown below demonstrate that our CV procedure provides reasonably stable and robust estimation of $ \epsilon $ even when the labeled sample size is small.
>
> |||$n=200$||||$n=500$||||$n=1000$|||
> |:---|:---|:---|:---|:---|:---|:---|:---|:---|:---|:---|:---|:---|
> |Method||Error|Coverage|Volume||Error|Coverage|Volume||Error|Coverage|Volume|
> |PPI|Original|10.315|0.000|53.459||10.315|0.000|13.772||10.314|0.000|4.863|
> ||Roica|0.131|0.884|0.036||0.085|0.864|0.009||0.058|0.918|0.003|
> ||Roica-cv|0.145|0.838|0.040||0.093|0.848|0.011||0.064|0.880|0.004|
> |PPI++|Original|>100|0.000|>100||>100|0.000|>100||>100|0.000|>100|
> ||Roica|0.132|0.884|0.035||0.086|0.870|0.009||0.058|0.914|0.003|
> ||Roica-cv|0.148|0.842|0.041||0.094|0.840|0.011||0.064|0.896|0.004|
> |PDC|Original|>100|0.000|>100||>100|0.000|>100||>100|0.000|>100|
> ||Roica|0.138|0.842|0.033||0.089|0.848|0.009||0.067|0.868|0.003|
> ||Roica-cv|0.153|0.800|0.040||0.095|0.826|0.011||0.067|0.878|0.004|
> |RePPI|Original|>100|0.000|>100||>100|0.000|>100||>100|0.000|>100|
> ||Roica|0.126|0.838|0.024||0.085|0.814|0.007||0.061|0.892|0.003|
> ||Roica-cv|0.145|0.810|0.034||0.093|0.800|0.009||0.062|0.882|0.003|
> |Supervised||0.177|0.842|0.073||0.112|0.860|0.020||0.076|0.892|0.007|
>
>
> > _The theoretical results assume mild contamination. Can the method tolerate higher contamination levels, and how would it perform when this assumption fails?_
>
> We thank the reviewer for this important question regarding the practical limits of our method beyond its theoretical assumptions. We have tried our method with larger values of $\epsilon$ (specifically, 0.2, 0.3, and 0.4). The empirical results of linear regression presented in our response to Reviewer qvSq and are included in Appendix C.4 of the revised manuscript.  All findings confirm that the method empirically tolerates these higher levels, maintaining good coverage and efficiency.
>
> > _How would Roica compare against robust domain adaptation or covariate-shift correction methods, which also address distributional heterogeneity?_
>
> We thank the reviewer for this comment. It is important to clarify that Roica is not directly comparable to robust domain adaptation methods, as they solve different problems. Domain adaptation methods, such as those using propensity score weighting, aim to correct for non-uniform sampling of covariates. This requires the sampling probabilities to be known or well-estimated. Our work primarily addresses the problem of outliers and contamination in the unlabeled data. The two frameworks are, in fact, complementary. A promising future direction would be to combine propensity score weighting with our robust calibration to handle datasets suffering from both covariate shift and data contamination, thereby achieving a higher level of robustness.

---

> > ### Comment · Reviewer_m9vm · 2025-11-26
> >
> > I thank the authors to address my questions. For synthetic data or simulation studies, one can always design in a comprehensive way to demonstrate the outperformance of the proposed method. However, why the proposed method is of interest in real empirical studies? Do you have concrete examples?
> >
> > In the current PPI literature, there are existing work that does not assume the homogeneity between labeled and unlabeled data, to address different aspects of distribution shift in different respects. It is not clear how this work is relevant/different from those existing work.

---

> > > ### Author Response · Authors · 2025-11-28
> > > **Response to Reviewer m9vm**
> > >
> > > We sincerely thank the reviewer for this insightful and constructive suggestion, which has greatly helped us improve the manuscript.
> > > > _For synthetic data or simulation studies, one can always design in a comprehensive way to demonstrate the outperformance of the proposed method. However, why the proposed method is of interest in real empirical studies? Do you have concrete examples?_
> > > >
> > >
> > > This is a valuable suggestion. Applying our method to real data with pre-existing data corruption, rather than solely relying on synthetic contamination, is a compelling next step. We will follow this direction in our subsequent research.
> > >
> > > >_In the current PPI literature, there are existing work that does not assume the homogeneity between labeled and unlabeled data, to address different aspects of distribution shift in different respects. It is not clear how this work is relevant/different from those existing work._
> > > >
> > >
> > > As noted in our previous response, our work addresses a distinct problem from methods handling covariate shift. While domain adaptation uses propensity score weighting to correct for covariate shift, it requires known sampling probabilities. Our work specifically targets data contamination (outliers) in the unlabeled set. The methods are complementary; future work could combine these approaches for comprehensive robustness. We also include a discussion clarifying this distinction and the potential for combining two approaches in the revised manuscript (highlighted in blue).

---

### Official Review · Reviewer_qvSq · 2025-11-01

**Soundness:** 3
**Presentation:** 3
**Contribution:** 3
**Rating:** 6
**Confidence:** 3

**Summary:**

The author focuses on the issue of estimating the overall data distribution parameter θ in semi-supervised scenarios, and how to improve the precision of θ estimation when there is distribution shift or data corruption in the unlabeled (or unreliable labeled) data. The author designed a robust parameter θ estimation technique Roica based on current PPI technology. By cross-validation to estimate the data contamination rate ϵ, with a weight restriction strategy on the unlabeled sample, the negative impact of bad data on the parameter θ estimation process was reduced. Detailed mathematical proofs and abundant simulations and real data experiments demonstrate the effectiveness of the proposed method.

**Strengths:**

1. Distribution shift/contamination in unlabeled data is widespread in real-world scenarios, but in previous studies of the PPI problems, this issue did not receive sufficient attention. The author pointed out this phenomenon for the first time and provided reasonable solutions.
2. Compared with label correction or fine-tuning of pretraining models commonly used in semi-supervised learning, Roica can directly estimate parameter θ without the similar operations, which means lower computational overhead of this algorithm.
3. Roica is a pluggable PPI method, which contains rigorous mathematical proofs. Rich experimental results have verified the effectiveness.

**Weaknesses:**

1. The unlabeled data contamination rate ϵ in the experiment is relatively low (<15%), making it impossible to confirm the robustness of the proposed method in scenarios with high contamination rate ϵ.
2. The simulation data and real dataset scenarios adopted are relatively simple. Whether it can ensure effectiveness in complex CV, NLP, and TS (time series) data machine learning tasks remains to be further confirmed.
3. The detailed introduction of the Roica and Roica-cv method settings in the experiments were not found, which might cause readers to be confused about the experimental results.

**Questions:**

1. Encourage the author to try to increase ϵ for experimental testing in the more complex dataset. In many machine learning tasks, there is often a distribution shift between all unlabeled samples and labeled samples (transfer learning), or the noise rate in noisy labeled tasks exceeds 40% (noisy label learning). The dataset used is also a more complex task rather than such logistic regression task.
2. I didn't find the author's explanation regarding the differences between Roica and RoicA-CV in the main text. I guess RoICA-CV is the experimental result of estimating ϵ using cross-validation. Then, is the ϵ in Roica use the real ϵ directly in the dataset? Otherwise, it is difficult to explain why Roica performs better than RoicA-CV. It is hoped that the author will provide necessary explanations here in the experimental section of the main text.
3. Although the paper provides a detailed description of the experimental setup, it is still recommended that the source code be published after this paper is accepted to promote further development in the field of PPI research.
4.How exactly is the weight vector w computed in experiments? Is it via convex optimization, filtering, or heuristic truncation?
5.It is recommended that all formulae be numbered.

---

> ### Author Response · Authors · 2025-11-21
> **Response to Reviewer qvSq**
>
> We appreciate the reviewer's constructive comments and have provided a point-by-point response to each, as detailed below. (The original comments are included in italics.)
>
> > _Encourage the author to try to increase $\epsilon$ for experimental testing in the more complex dataset. In many machine learning tasks, there is often a distribution shift between all unlabeled samples and labeled samples (transfer learning), or the noise rate in noisy labeled tasks exceeds 0.4 (noisy label learning). The dataset used is also a more complex task rather than such logistic regression task._
>
> 1. Following the reviewer's suggestion, we evaluated our method using larger values of $\epsilon$ (specifically, 0.2, 0.3, and 0.4). The results for linear regression are presented in the table below and have been included in Appendix C.4 of the revised manuscript. These findings confirm that our method maintains robust performance across a wide range of noise levels, even at $\epsilon = 0.4$.
> 2. More complex tasks/models:  The PPI framework is primarily designed to leverage modern prediction tools, such as neural network-based methods, alongside large unlabeled datasets to enhance statistical inference and hypothesis testing. For instance, in the real-data example in Section 6.2, we employed a pre-trained transformer model as the base predictor. We agree with the reviewer that incorporating widely recognized benchmarks, such as CIFAR and MNIST, would further strengthen the breadth of this work.
>
> | | | $\epsilon = 0.2$ | | | | $\epsilon = 0.3$ | | | | $\epsilon = 0.4$ | | |
> |:---|:---|:---|:---|:---|:---|:---|:---|:---|:---|:---|:---|:---|
> | Method | | Error | Coverage | Volume | | Error | Coverage | Volume | | Error | Coverage | Volume |
> | PPI | Original | 10.353 | 0.000 | 4.944 | | 10.360 | 0.000 | 4.979 | | 10.363 | 0.000 | 5.003 |
> | | Roica | 0.058 | 0.906 | 0.003 | | 0.059 | 0.904 | 0.003 | | 0.060 | 0.904 | 0.003 |
> | | Roica-cv | 0.063 | 0.886 | 0.004 | | 0.065 | 0.880 | 0.004 | | 0.066 | 0.874 | 0.004 |
> | PPI++ | Original | >100 | 0.000 | >100 | | >100 | 0.000 | >100 | | >100 | 0.000 | >100 |
> | | Roica | 0.058 | 0.910 | 0.003 | | 0.059 | 0.910 | 0.003 | | 0.060 | 0.904 | 0.003 |
> | | Roica-cv | 0.065 | 0.884 | 0.004 | | 0.063 | 0.884 | 0.004 | | 0.065 | 0.888 | 0.004 |
> | PDC | Original | >100 | 0.000 | >100 | | >100 | 0.000 | >100 | | >100 | 0.000 | >100 |
> | | Roica | 0.067 | 0.874 | 0.003 | | 0.062 | 0.896 | 0.003 | | 0.075 | 0.876 | 0.004 |
> | | Roica-cv | 0.067 | 0.870 | 0.004 | | 0.066 | 0.884 | 0.004 | | 0.076 | 0.866 | 0.004 |
> | RePPI | Original | >100 | 0.000 | >100 | | >100 | 0.000 | >100 | | >100 | 0.000 | >100 |
> | | Roica | 0.059 | 0.890 | 0.003 | | 0.063 | 0.874 | 0.003 | | 0.089 | 0.852 | 0.004 |
> | | Roica-cv | 0.063 | 0.878 | 0.003 | | 0.065 | 0.864 | 0.004 | | 0.066 | 0.856 | 0.004 |
> | Supervised | | 0.076 | 0.892 | 0.007 | | 0.076 | 0.892 | 0.007 | | 0.076 | 0.892 | 0.007 |
>
>
> > _I didn't find the author's explanation regarding the differences between Roica and RoicA-CV in the main text ...... It is hoped that the author will provide necessary explanations here in the experimental section of the main text._
> >
>
> We appreciate the reviewer's insightful observation. The intuition is correct: the distinction between **Roica** and **Roica-cv** lies in how the parameter $\epsilon$ is determined. **Roica** operates in an 'oracle' setting using the ground-truth $\epsilon$ from the dataset, whereas **Roica-cv** estimates $\epsilon$ via cross-validation. We have refined the discussion of the cross-validation procedure in Section 3.3 and updated the experimental section of the revised manuscript to explicitly clarify this distinction.
>
>
>
> > _Although the paper provides a detailed description of the experimental setup, it is still recommended that the source code be published after this paper is accepted to promote further development in the field of PPI research._
> >
>
> We agree with the reviewer on the importance of code availability. The source code is prepared and will be made public upon the paper's acceptance.
>
>
>
> > _How exactly is the weight vector w computed in experiments? Is it via convex optimization, Filtering, or heuristic truncation?_
> >
>
> We appreciate the reviewer's question about the computation of the weight vector **w**. This is indeed computed through the Filtering method [1]. We chose this approach for its dimension-agnostic robustness and efficient computation. The complete algorithmic steps are added in Algorithm 1 in Appendix D.1 of the revised manuscript.
>
> > _It is recommended that all formulae be numbered._
> >
>
> We thank the reviewer for this suggestion. We have numbered the most important equations in the revised manuscript to improve readability and facilitate reference.
>
>
>
> [1] Diakonikolas, I., Kamath, G., Kane, D.M., Li, J., Moitra, A. and Stewart, A., 2017, July. Being robust (in high dimensions) can be practical. In International Conference on Machine Learning (pp. 999-1008). PMLR.

---

### Note · Program_Chairs · 2026-01-17
**Submission Desk Rejected by Program Chairs**

The following references in this submission do not refer to real documents and/or have major errors in bibliographic information:

 Guanbin Li, Yipin Zhang, Xiangyang Li, Yifan Sun, Zhe Wang, Hang Xu, Tian Chen, and Yizhou Zhang. Adversarial attacks on principal component analysis. IEEE Transactions on Information Forensics and Security, 17:2475-2487, 2022.